# Proteolysis-targeting chimera against BCL-X$_L$ destroys tumor-infiltrating regulatory T cells

Ryan Kolb[1,2,11], Umasankar De[1,11], Sajid Khan [3], Yuewan Luo[1], Myung-Chul Kim[1], Haijun Yu[1], Chaoyan Wu[1], Jiao Mo[1], Xin Zhang[3], Peiyi Zhang[4], Xuan Zhang [4], Nicholas Borcherding[5], Daniel Koppel[2,6], Yang-Xin Fu [7], Song Guo Zheng[8], Dorina Avram[2,9,10], Guangrong Zheng [2,4], Daohong Zhou [2,3,12✉] & Weizhou Zhang [1,2,12✉]

Regulatory T cells (Tregs) play an important role in maintaining immune homeostasis and, within tumors, their upregulation is common and promotes an immunosuppressive micro-environment. Therapeutic strategies that can eliminate Tregs in the tumor (i.e., therapies that do not run the risk of affecting normal tissues), are urgently needed for the development of cancer immunotherapies. Here we report our discovery of B-cell lymphoma extra-large (BCL-X$_L$) as a potential molecular target of tumor-infiltrating (TI) Tregs. We show that pharmacological degradation of BCL-X$_L$ using a newly developed platelet-sparing BCL-X$_L$ Proteolysis-targeting chimera (PROTAC) induces the apoptosis of TI-Tregs and the activation of TI-CD8$^+$ T cells. Moreover, these activities result in an effective suppression of syngeneic tumor growth in immunocompetent, but not in immunodeficient or CD8$^+$ T cell-depleted mice. Notably, treatment with BCL-X$_L$ PROTAC does not cause detectable damage within several normal tissues or thrombocytopenia. These findings identify BCL-X$_L$ as a target in the elimination of TI-Tregs as a component of cancer immunotherapies, and that the BCL-X$_L$-specific PROTAC has the potential to be developed as a therapeutic for cancer immunotherapy.

[1] Department of Pathology, Immunology, and Laboratory Medicine, College of Medicine, University of Florida, Gainesville, FL, USA. [2] University of Florida Health Cancer Center, University of Florida, Gainesville, FL, USA. [3] Department of Pharmacodynamics, College of Pharmacy, University of Florida, Gainesville, FL, USA. [4] Department of Medicinal Chemistry, College of Pharmacy, University of Florida, Gainesville, FL, USA. [5] Department of Pathology, University of Iowa, Iowa City, IA, USA. [6] Department of Chemistry, College of Liberal Art and Sciences, University of Florida, Gainesville, FL, USA. [7] Department of Pathology, University of Texas Southwestern Medical Center, Dallas, TX, USA. [8] Department of Internal Medicine, Ohio State University College of Medicine and Wexner Medical Center, Columbus, OH, USA. [9] Department of Anatomy and Cell Biology, College of Medicine, University of Florida, Gainesville, FL, USA. [10] Department of Immunology, Moffitt Cancer Center, Tampa, FL, USA. [11] These authors contributed equally: Ryan Kolb, Umasankar De. [12] These authors jointly supervised this work: Daohong Zhou, Weizhou Zhang. ✉email: zhoudaohong@cop.ufl.edu; zhangw@ufl.edu

A growing body of evidence demonstrates that regulatory T cells (Tregs) play an important role in cancer progression and that they do so by suppressing cancer-directed immune responses[1]. Tregs have been targeted for destruction by exploiting antibodies against and small-molecule inhibitors of several molecules that are highly expressed in Tregs —including immune checkpoint molecules, chemokine receptors, and metabolites[2,3]. To date, these strategies have had only limited antitumor efficacy; yet they have also created significant risk of autoimmunity because most of them do not differentiate Tregs in tumors from those in normal tissues.

We have explored Treg targeting from a different angle, determining how tumor-infiltrating (TI) Tregs survive the adverse tumor-immunosuppressive microenvironment, which is characterized by a stress condition resulting from the low supply of oxygen and nutrients and a high concentration of metabolic waste[4-6]. The pro-apoptotic nature of such environments can be countered by anti-apoptotic proteins of the BCL-2 family, such as BCL-2, B-cell lymphoma extra-large (BCL-$X_L$), or MCL-1. These proteins have been extensively targeted with small-molecule inhibitors in cancer therapies[7,8], including BCL-2- and BCL-$X_L$-dual inhibitor ABT263 (navitoclax)[9], BCL-2 inhibitor ABT199 (venetoclax)[10], BCL-$X_L$ inhibitor A-1331852 (ref. [11]), and MCL-1 inhibitor S63845 (ref. [12]). ABT199, the BCL-2 specific inhibitor, has been approved by the FDA for cancer therapy[13]. The inhibition of BCL-$X_L$ with an inhibitor, however, induces severe thrombocytopenia, an on-target and dose-limiting toxicity, which limits the use of ABT263 and other BCL-$X_L$ inhibitors as safe and effective anticancer drugs in clinic[14,15].

To avoid the on-target toxicity of BCL-$X_L$ inhibitors to platelets, we have recently reported the development of the first-in-class BCL-$X_L$ degrader—a proteolysis-targeting chimera (PROTAC) referred to as DT2216—targets BCL-$X_L$ to the von Hippel-Lindau (VHL) E3 ligase for ubiquitination and subsequent proteasomal degradation[16]. Because platelets express a low level of VHL E3 ligase, DT2216 has minimal effect on platelets but exhibits an improved cytotoxicity against various cancer cells that are dependent on BCL-$X_L$ for survival[16,17]. We also developed another similar PROTAC PZ15227 that uses cereblon (CRBN) E3 ligase to induce BCL-$X_L$ polyubiquitination and degradation[18].

Here we show that BCL-$X_L$ is highly expressed within the TI-Treg population from renal cell carcinoma (RCC) and several other human cancers. Using the two PROTACs (DT2166 and PZ15227) to degrade BCL-$X_L$, we are able to define the critical pro-survival function of BCL-$X_L$ within TI-Tregs, which represents a potential method to deplete TI-Tregs for cancer immunotherapy.

## Results

**Profiling of BCL-2 family proteins in immune cells.** We recently developed a single-cell RNA sequencing (scRNAseq) dataset including a total of 12,239 TI immune cells and 13,433 peripheral blood (PB) mononuclear cells (paired PB immune cells from the same patients) from three human clear cell renal cell carcinomas (ccRCCs). A total of 22 clusters of immune cells were annotated for different immune cell types based on the t-distributed stochastic neighbor embedding (tSNE) machine-learning algorithm (Supplementary Fig. 1a) with defined T cell subtypes based on various markers (Supplementary Fig. 1b). In TI-Tregs within ccRCCs, the expression of BCL2L1 (the gene encoding BCL-$X_L$) was higher than that in PB-Tregs and other T cells (CD4+ conventional T cells (Tconv) and CD8+ T cells) (Fig. 1a and Supplementary Fig. 1c, shown as Napierian log fold-change versus percent difference compared to PB-counterparts). The expression of BCL2L1 was also high in TI-Tregs in human

hepatocellular carcinoma (HCC scRNAseq dataset GSE98638)[19] (Supplementary Fig. 1d, shown as Napierian log fold-change versus percent difference compared to PB-counterparts), as well as in mouse MC38 colon adenocarcinoma (Fig. 1b and Supplementary Table 1, here the comparison is to spleen Tregs shown as log2 (n + 1)-transformed data). Other BCL2 family genes such as PMAIP1 (NOXA) and BAX were also higher in TI-Tregs than other PB-Tregs cells from human ccRCC (Supplementary Fig. 1c); MCL1 was slightly higher in TI-Tregs than PB-Tregs in HCC (Supplementary Fig. 1d). The expression of genes such as BCL2 did not differ significantly between TI-Tregs and other TI-T cells (Fig. 1a).

An analysis of published RNAseq datasets from previous reports[20,21] revealed that BCL2L1 levels were elevated in TI-Tregs across many cancer types and species in comparison to Tregs from normal tissues and other TI-T cell types (Supplementary Fig. 1e). The cancer tissues included human breast (BrCa) and colorectal (CRC) cancers, as well as mouse CRC cancers (CT26 and MC38) and melanomas (B16 and BP) (Supplementary Fig. 1e). Tregs from normal tissues (colon, pancreas, muscle, fat) expressed lower levels of BCL2L1 than counterparts from spleen (Supplementary Fig. 1e). Using the reverse graph embedding approach in the Monocle 2 algorithm[22], we constructed a cell trajectory of the TI-Tregs and PB-Tregs in the ccRCC (Supplementary Fig. 1f). The trajectory from PB to TI Tregs contained two distinct branches (Supplementary Fig. 1f) that were not detectable using marker genes nor revealed in our previous tSNE clustering analysis (Supplementary Fig. 1a). BCL2L1—but not BCL2—was significantly upregulated within the branch 1 (Cell Fate #1) TI-Tregs relative to Cell Fate #2 TI-Tregs or PB-Tregs (Supplementary Fig. 1g); whereas both branches expressed Treg markers FOXP3 and IL2RA (CD25) (Supplementary Fig. 1h). Other immune-suppressive molecules that are known to mediate Treg function, such as ENTPD1 (gene encoding CD39) (Supplementary Fig. 1g), are also elevated along the Cell Fate #1; however, another gene that is also critical to convert ATP to AMP, i.e. NT5E (gene encoding CD73), is not detectable in TI-Tregs (Supplementary Fig. 1g). Using flow cytometry of human cancer specimens (Supplementary Fig. 2a), we examined the heterogeneity of human TI-Tregs using effector/memory markers CD45RA and CCR7. Although PB-Tregs exhibited subpopulations of naïve (CD45RA+CCR7+), effector (EMRA, CD45RA+ CCR7−), central memory (CM, CD45RA−CCR7+), and effector memory (EM, CD45RA−CCR7−), TI-Tregs only consisted of the EM population (Supplementary Fig. 2b), which is consistent with the observation from Sakaguchi laboratory who initially developed the use of CD45RA for human Treg heterogeneity[23].

Using flow cytometry, we found that the levels of BCL-$X_L$ protein—determined by intracellular staining of BCL-$X_L$—were much higher in TI-Tregs versus PB-Tregs in both human BrCa (Fig. 1c, d) and ccRCC (Fig. 1e, f). BCL-$X_L$ levels were higher in TI-Tregs than those within PB or TI CD8+ and Tconv cells in human BrCa (Fig. 1d and Supplementary Fig. 3a) and ccRCC (Fig. 1e, f and Supplementary Fig. 3b). BCL-$X_L$ levels were also higher, though to a lesser extent, in conventional TI-CD4+ Tconv versus their PB-counterparts (Fig. 1d, f); whereas TI-CD8+ T cells exhibited increased BCL-$X_L$ in BrCa (Fig. 1d) but decreased level in ccRCC relative to PB CD8+ T cells (Fig. 1f). A similar result was observed in mouse Py8119 BrCa tumors (Supplementary Fig. 3c). BCL-$X_L$ levels were also significantly higher in TI-Tregs with all human cancer specimens combined, including the nine BrCa and paired PBs, two ccRCC and paired PBs, and two colon cancer and paired PBs (Supplementary Fig. 3d). BCL-$X_L$ levels were similar in PB-CD8+, PB-Tconv, and PB-Treg from same BrCa patients (Fig. 1d and Supplementary Fig. 3d). As the major anti-cancer immune cell types, PB- and TI-CD8 T cells exhibited

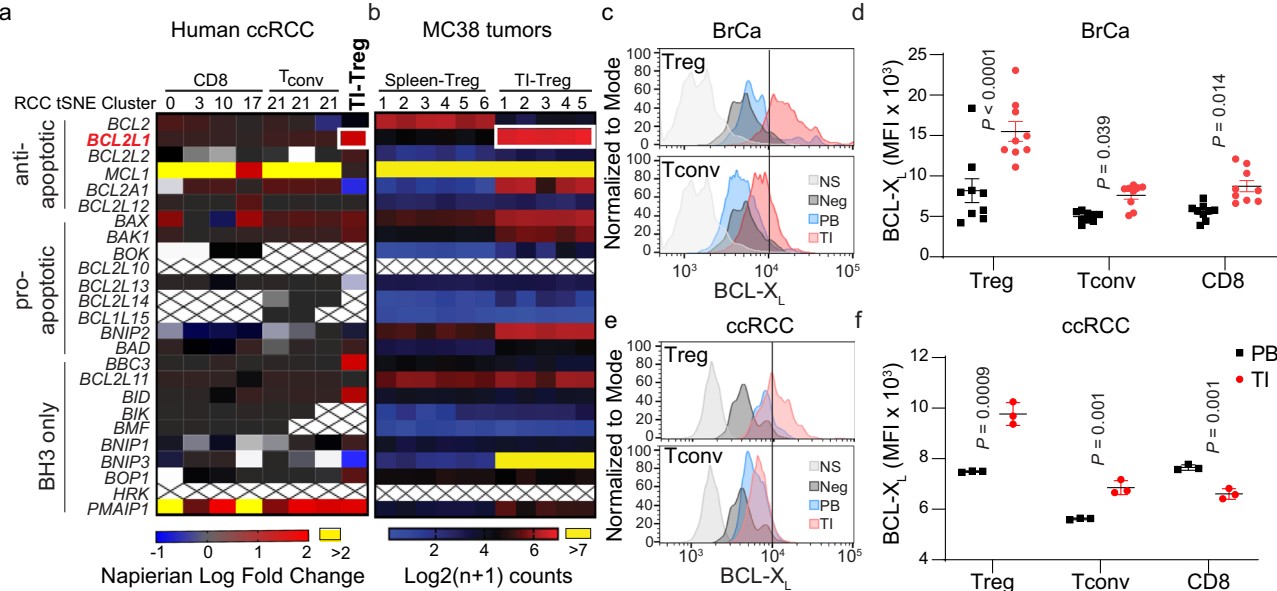

**Fig. 1 BCL-X$_L$ is selectively upregulated in TI-Tregs. a** Heatmap depicting the fold difference in expression of members of the BCL-2 gene family from distinct subsets of TI-T cells relative to T cell counterparts from peripheral bloods (PB) of paired RCC patients. TI-T cells include CD8$^+$, conventional CD4$^+$ (Tconv), and TI-Tregs. See Supplementary Fig. 1a for characterization of immune cell clusters by tSNE plot of scRNAseq data. Cross-hatching indicates no expression. **b** Heatmap depicting the expression of BCL-2 family genes in mouse MC38 colon adenocarcinoma TI-Tregs and splenic Tregs. **c** Representative histogram showing BCL-X$_L$ levels in PB- and TI-Tregs or TI-Tconv from human breast cancer (BrCa), and **d** median fluorescence intensity (MFI) of BCL-X$_L$ staining ± s.d. in PB- or TI-Tregs, CD8$^+$, and Tconv in these cancers ($n = 9$ biological replicates). **e** Representative histogram showing BCL-X$_L$ levels in PB- and TI-Tregs or TI-Tconv from human renal cancer and paired blood, and **f** MFI of BCL-X$_L$ staining ± s.d. in PB- or TI-Tregs, CD8$^+$, and Tconv in these cancers ($n = 3$). Unpaired T-tests (**d**, $n = 9$; and **f**, $n = 3$ biological replicates, two-sided) were performed and the corresponding P values are indicated. **c, e** NS, non-staining; Neg, negative control, treated with DT-2216 to deplete BCL-X$_L$ as negative control.

different populations of effector/memory phenotypes (Supplementary Figs. 2b and 3e, f), all of which exhibited lower expression of BCL-X$_L$ relative to TI-Tregs (Supplementary Fig. 3g, h). The selective upregulation of BCL-X$_L$ in TI-Tregs, as well as high levels of the pro-apoptotic BH3$^-$only proteins NOXA (encoded by *PMAIP1*), BID and BAX (Fig. 1a), which inhibit MCL-1 and BCL-2 respectively, suggest that TI-Tregs may primarily depend on BCL-X$_L$ for survival[24].

**Targeting BCL-X$_L$ for degradation in TI-Tregs.** To test whether BCL-X$_L$ is required for the survival of TI-Tregs, we used our newly generated PROTACs to target BCL-X$_L$. DT2216, the lead BCL-X$_L$ PROTAC, targets BCL-X$_L$ to the VHL E3 ligase for polyubiquitination and degradation[16]. DT2216 did not cause the platelet toxicity observed when other BCL-X$_L$ inhibitors were used to inhibit BCL-X$_L$[25], because of the poor expression of VHL in platelets. To assess the effectiveness of PROTAC-mediated BCL-X$_L$ degradation, we developed an ex vivo cancer slice culture system (Supplementary Fig. 4a). Treatment with DT2216 led to a significant decrease in BCL-X$_L$ levels within TI-Tregs from human BrCa (Fig. 2a, b and Supplementary Fig. 4b, c) and RCC (Fig. 2c, d and Supplementary Fig. 4d). Whereas the treatment also reduced BCL-X$_L$ levels in TI-Tconv (CD4$^+$ FOXP3$^-$ T cells) and TI-CD8$^+$ T cells, this effect was more modest (Fig. 2b, d and Supplementary Fig. 4b–d). Similar results were observed in cultures of MC38 colon adenocarcinoma slices (Fig. 2e, f).

To study the role of BCL-X$_L$ in the tumor immune microenvironment, we chose several cell lines with which to produce syngeneic tumor models (MC38 for colon adenocarcinoma, 4T1 and Py8119 for breast cancer, and Renca for renal cancer). These cell lines express various levels of VHL, CRBN, BCL-X$_L$, BCL-2, and MCL-1 (Supplementary Figs. 4e and 12). The DT2216 (ref. [16]) and PZ15227 (a second BCL-X$_L$ PROTAC

that targets BCL-X$_L$ to the CRBN E3 ligase for polyubiquitination and degradation)[18] treatments each resulted in dose-dependent reduction of BCL-X$_L$ levels (Fig. 2g and Supplementary Figs. 4f, 13, and 14). Treatment of mice bearing tumors originated from MC38 and Renca cells with DT2216 resulted in reduced BCL-X$_L$ levels in the tumors, regardless of whether the mice were wild-type (WT) (Fig. 2h and Supplementary Fig. 13) or immunodeficient (NOD-SCID and null for IL-2 receptor gamma (NSG)) (Supplementary Figs. 4g and 14).

**Degradation of BCL-X$_L$ within TI-Tregs induces anti-tumor immunity.** As we have seen efficient degradation of BCL-X$_L$ by its PROTACs, we first checked whether the mouse cell lines are sensitive to BCL-X$_L$ PROTACs. We found that in vitro treatment of MC38, Renca, Py8119, or 4T1 cells with DT2216 or PZ15227 had little effect on cell survival (Supplementary Fig. 5a, b). This is because these cell lines also express the anti-apoptotic proteins BCL-2 and MCL-1 (Supplementary Fig. 4e), and thus are not solely dependent on BCL-X$_L$ for survival, as confirmed by a BH3-pairing assay using inhibitors specific for BCL-2, BCL-X$_L$, and MCL-1 (Supplementary Fig. 5c–f). BCL-X$_L$ deficiency, caused by either treatment with DT2216 or genetic knockout (KO), influenced neither apoptosis nor colony-forming efficiency of cancer cells in vitro (Supplementary Fig. 5g–i).

Although BCL-X$_L$ PROTACs did not kill the above cancer cells in vitro, DT2216 treatment resulted in significantly reduced tumor growth of the tested models—including Renca (Fig. 3a), MC38 (Fig. 3b), and Py8119 (Fig. 3c) tumors in WT immunocompetent mice. A similar result was observed in the Renca model after treatment with PZ15227 (ref. [18]) (Fig. 3a). Notably, CRISPR/Cas9-mediated KO of BCL-X$_L$ in Renca cells resulted in a much smaller reduction in tumor growth (Supplementary Figs. 6a and 15). Also, DT2216 lost its tumor-

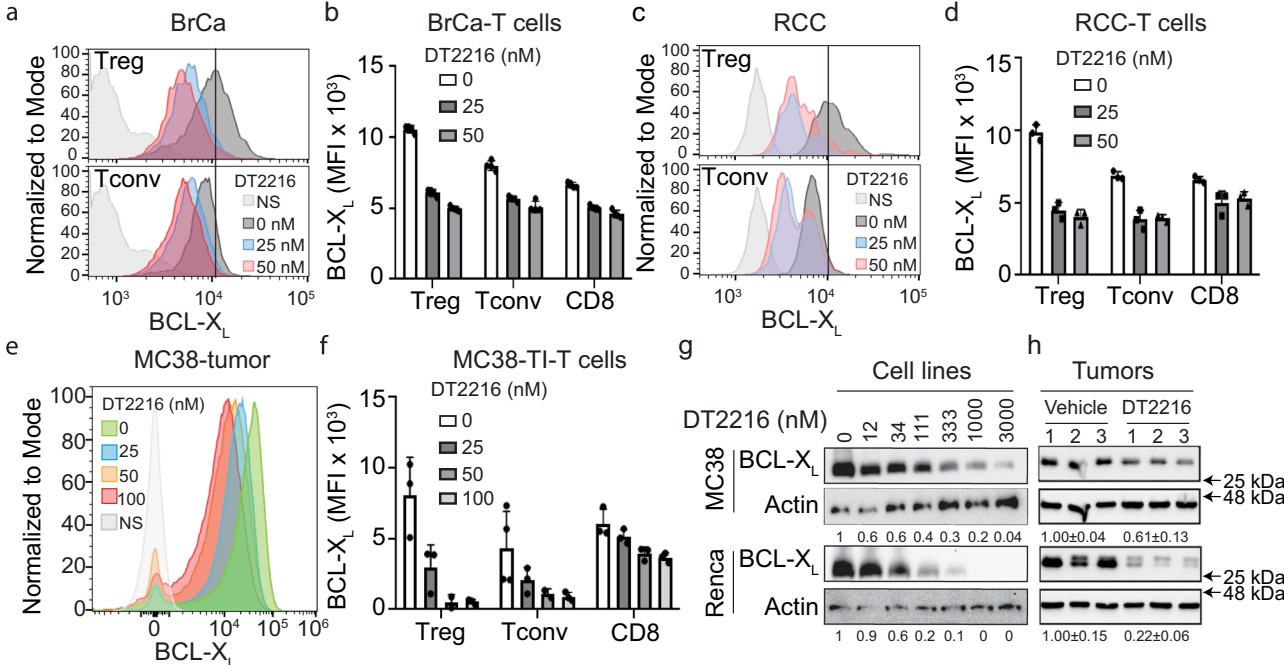

**Fig. 2 BCL-XL PROTAC leads to efficient degradation of BCL-X_L in the tumor microenvironment. a–d** BCL-X_L levels in TI-Tregs or Tconv in ex vivo slice cultures of human cancer specimens, including **a, b** breast cancer and **c, d** renal cancer, following treatment with DT2216 for 48 h. Representative histograms showing BCL-X_L levels in TI-Tregs or TI-Tconvs from human **a** breast cancer and **c** renal cancer; **b, d** MFI of BCL-X_L in the indicated TI-T cells ± s.d. ($n = 4$ biological replicates). **e, f** BCL-X_L levels in TI-Tregs in ex vivo slice cultures from MC38 colon adenocarcinoma following treatment with the DT2216 for 48 h. **e** Representative histogram showing BCL-X_L levels in TI-Tregs and **f** MFI of BCL-X_L in the indicated T cells ± s.d. (**f**, $n = 3$ biological replicates). **g** BCL-X_L expression in murine cancer cell lines including MC38 and Renca, following treatment with DT2216 ($n = 2$ biological replicates). **h** BCL-X_L expression in murine tumors, including MC38 and Renca tumors, following in vivo treatment with DT2216 ($n = 3$ biological replicates, mean ± s.d.).

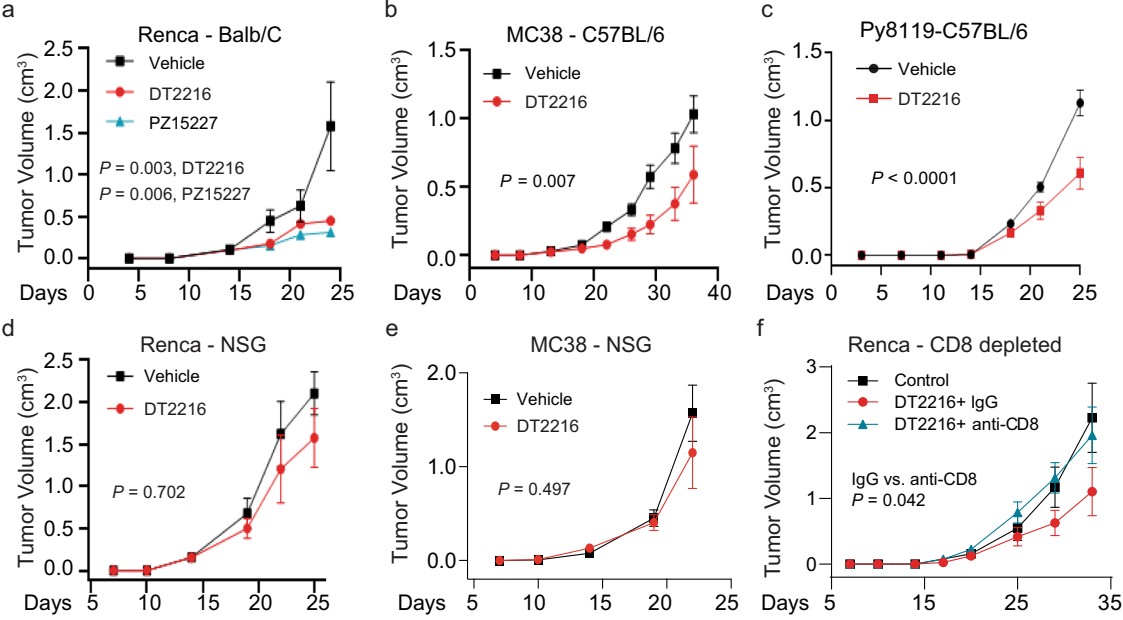

**Fig. 3 Depleting the tumor microenvironment of BCL-X_L reduces tumor growth in a CD8⁺ T cell-dependent manner. a–c** Effect of treatment with DT2216 on tumor growth from syngeneic immunocompetent mouse models. **a** Renca tumor volume ± s.e.m in Balb/C mice ($n = 6$ biological replicates) treated with DT2216, PZ15227, or vehicle; **b** MC38 tumor volume ± s.e.m. in C57BL/6j mice ($n = 10$ biological replicates) treated with DT2216 or vehicle; and **c** Py8119 tumor volume ± s.e.m. in C57BL/6j female mice ($n = 6$ biological replicates) treated with DT2216 or vehicle. **d, e** Effect of treatment with DT2216 on tumor growth from immunodeficient NSG model. **d** Renca ($n = 6$ biological replicates) or **e** MC38 ($n = 10$ biological replicates) tumor volume ± s.e.m in NSG mice treated with vehicle or DT2216. **f** Graph depicting Renca tumor volume ± s.e.m. in Balb/C mice ($n = 8$ biological replicates) following antibody-mediated depletion of CD8 (see Supplementary Fig. 6b) or control (IgG), and treated with DT2216 similarly as in **a**. A two-sided two-way ANOVA was performed for all tumor growth curves with P values for the interaction between the curves indicated.

inhibitory effect in the same tumor cell lines when injected into NSG mice, which lack T and B lymphocytes, as well as natural killer cells (Fig. 3d, e). Collectively, our results suggest that DT2216 exerts its antitumor activity via a mechanism that is dependent on immune cells. We found that depletion of CD8$^+$ T cells abrogated the DT2216-mediated tumor inhibitory effect (Fig. 3f and Supplementary Fig. 6b), supporting the notion that DT2216-mediated inhibition of tumor growth in syngeneic models is via CD8$^+$ T cells.

**Degradation of BCL-X$_L$ leads to the reduction of TI-Tregs and activation of CD8$^+$ T cells.** To determine the extent to which BCL-X$_L$ depletion affects the immunosuppressive environment of tumors, we profiled lymphocytes of MC38 tumor-bearing mice in Fig. 3b by flow cytometry (Supplementary Fig. 7a). The percentages of lymphocytes in the spleen, draining lymph node (DLN), and tumors were assessed in mice that had been treated once weekly for 3 weeks with DT2216 after tumors were palpable. DT2216 treatment significantly reduced the frequency of TI-Tregs in both the DLN and tumors; it did so to a lesser extent in the spleen (Fig. 4a). To determine whether this DT2216-mediated Treg depletion was due to increased apoptosis, we injected MC38 tumor-bearing mice with a fluorescent probe that detects active caspases. A significant increase in median fluorescence intensity (MFI) was observed in TI-Tregs upon DT2216 treatment (Fig. 4b and Supplementary Fig. 7b, c), indicative of active caspases. DT2216 treatment did not significantly affect the frequency of TI, PB, and splenic Tconv cells (Fig. 4c) or that of TI-Tconv and TI-CD11B$^+$ myeloid cells in which caspases were active (Fig. 4d and Supplementary Fig. 7b, d, e). Splenic cells, including Treg, Tconv,

CD8$^+$ T, and CD11B$^+$ myeloid cells, however, exhibited minimal caspase activation with or without DT2216 treatment (Supplementary Fig. 7f). CD39$^+$ TI-Tregs—shown to have suppressive capacity when undergoing apoptosis[26]—exhibited similar levels of cell death after DT2216 treatment (Supplementary Fig. 7g).

Although DT2216 had a small effect on the frequency on total CD8$^+$ T cells (Fig. 4e), a significant increase in the activation of these cells was indicated by the presence of granzyme B (Gzmb)$^+$ and perforin$^+$ CD8$^+$ T cells (Fig. 4f, g and Supplementary Fig. 7h) without significant change of Ki-67$^+$ proliferative cells (Fig. 4g). Similar increase of Gzmb$^+$ CD8$^+$ T cells was also seen in blood and DLN from tumor-bearing mice, but not spleen and other non-draining lymph nodes (NDLN) after DT2216 treatment (Supplementary Fig. 7i). This finding supports the notion that treatment with DT2216 leads to an immune active tumor microenvironment due to the elimination of TI-Tregs, resulting in tumor-specific CD8 T cell activation as seen from tumor, blood, and DLNs.

TI-Treg depletion by DT2216 was confirmed in ex vivo cancer slice cultures generated from mouse Py8119 BrCa (Supplementary Fig. 7j) or mouse Renca renal tumors (Supplementary Fig. 7k), as well as human BrCa (Fig. 4h) and CRC (Fig. 4i). Moreover, this immune-related tumor-inhibitory phenotype is specific to BCL-X$_L$ degradation, given that inhibition of BCL-2 with BCL-2-specific inhibitor ABT-199 (ref. [10]), which did not cause an increase in the activation of TI-CD8$^+$ T cells (Supplementary Fig. 7l), did not inhibit the growth of MC38 or Renca tumors, even with daily administration of much higher doses than DT2216 (Supplementary Fig. 7m, n). The impact of DT2216 was minimal to the majority of myeloid cell populations in tumor-bearing mice, including granulocytic or monocytic MDSC (CD11b$^+$Ly6G$^+$ G-MDSC or CD11b$^+$Ly6C$^+$Ly6G$^-$ M-

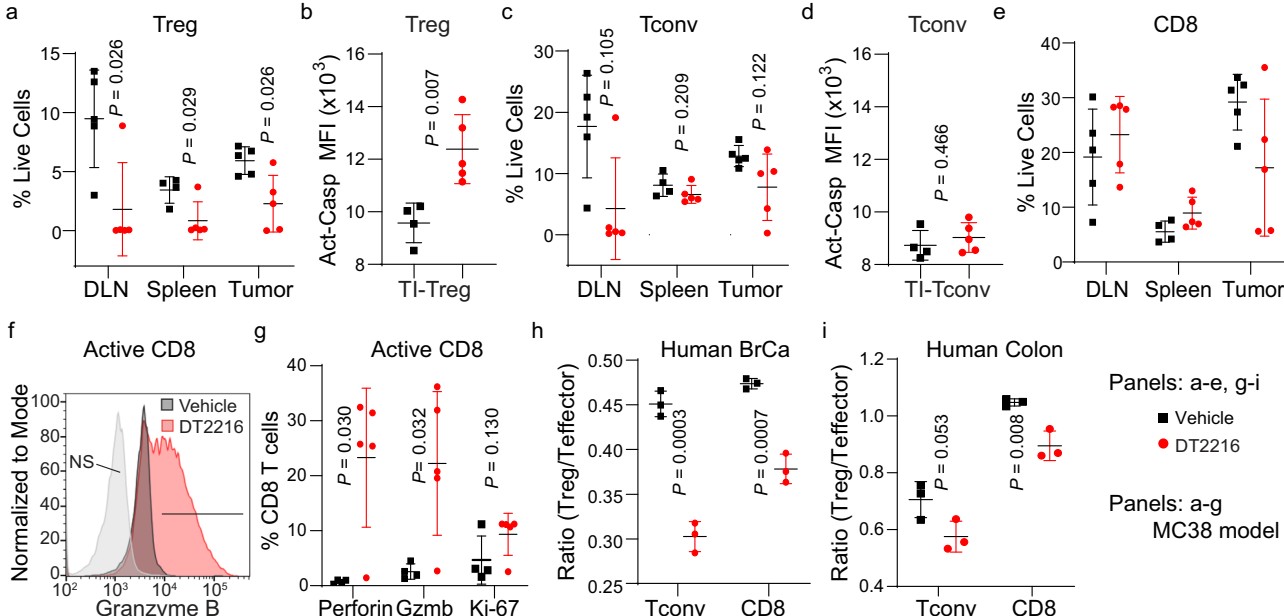

**Fig. 4 Depleting the tumor microenvironment of BCL-X$_L$ leads to an apoptosis-induced decrease in the number of TI-Tregs. a** Percentage of Tregs following treatment with the BCL-X$_L$ targeting PROTAC DT2216 (or vehicle) in the draining lymph nodes (DLN), spleens, and tumors from MC38 tumor-bearing mice. **b** Amount of pan-caspase labeling in TI-Tregs from MC38 tumor-bearing mice following treatment with DT2216 (or vehicle) as assessed by flow cytometry. MFI of pan-caspase activation probe is shown. **c–e** Percentages of Tconv (**c**) and CD8$^+$ T (**e**) cells following treatment with DT2216, and **d** amount of caspase labeling in TI-Tconv cells. **f** Representative histogram showing levels of granzyme B (Gzmb) in TI-CD8$^+$ T cells from MC38 tumor-bearing mice treated with DT2216 (or vehicle). **g** Number of perforin$^+$, granzyme B (Gzmb)$^+$, or Ki-67$^+$ TI-CD8$^+$ T cells from MC38 tumor-bearing mice following treatment with DT2216 (or vehicle); **a**, **c**, **e**, and **g**: $n = 5$ biological replicates for DLN and tumors, $n = 4$ biological replicates in control, and $n = 5$ in DT2216 group for spleen; **b** and **d**: $n = 4$ biological replicates in control and $n = 5$ in DT2216 group. **h**, **i** Ratios of TI-Treg/TI-Tconv or TI-Treg/TI-CD8$^+$ T cells in (**h**, $n = 3$ biological replicates) human breast cancer slice sections or (**i**, $n = 3$ biological replicates) in human colon cancer slices, after ex vivo treatment with DT2216 (25 nM, or vehicle). Shown are the mean ± s.d. Two-sided unpaired $t$ test was performed, with $P$ values indicated.

MDSC), macrophages (F4/80[+]), dendritic cells (CD11c[+]MHCII[+] DC), or CD8[+] DCs, with the exception of DC where DT02216 treatment led to slightly decreased TI-DC population (Supplementary Fig. 8a–d).

Treatment with DT2216 had no global effect on the overall levels of Treg, CD4[+] Tconv, CD8[+] T, NK cells, and B cells from various tissues within non-tumor bearing mice (Supplementary Fig. 9a–e), nor did we see significant difference in CD8 T cell memory and effector sub-populations from these tissues (Supplementary Fig. 9f–i), defined by CD45RA and CCR7 (Supplementary Fig. 2a, b). There was a slight increase of thymic Tregs (Supplementary Fig. 9a). These findings suggest that DT2216 may selectively deplete TI-Treg in an individual tumor/mouse-dependent manner.

Serum IgM and IgG levels were not altered by DT2216 treatment either from normal or tumor-bearing mice after an extended treatment of 5 weeks (Supplementary Fig. 9j, k), as compared with the elevated levels from pristane-treated mice (Supplementary Fig. 9j, k). PB-Tregs, -Tconv, or -CD8[+] T cells from non-tumor-bearing mice (Supplementary Fig. 9l, top), or total Tregs from the blood, lymph nodes, or spleens of non-tumor-bearing mice (Supplementary Fig. 9l, bottom) remained unaltered by DT2216. Histological analysis did not reveal tissue damage in the lung, pancreas, or intestines of Renca-tumor-bearing mice treated with either DT2216 or PZ15227 (Supplementary Fig. 9m), indicating that Treg depletion and CD8[+] T cell activation are relatively specific to the tumor microenvironment rather than being systemic and that targeting BCL-X$_L$ by PROTACs is a relatively safe and effective way to treat cancer. However, further studies are needed to examine whether DT2216 treatment causes any other autoimmune pathology in the organs such as liver, skin, and other organs as seen in *Foxp3*[−] mice[27] to ensure the safety of DT2216 before it can be translated into the clinic.

## Discussion

Our results reveal that targeting BCL-X$_L$ has the potential to improve cancer immunotherapy. This can be achieved pharmacologically by applying PROTACs that target BCL-X$_L$ for ubiquitination (and thereby for proteasome-mediated degradation) within the tumor microenvironment, but these PROTACs cannot eliminate platelets because the expression of VHL or CRBN in platelets is minimal[16,18]. Degradation of BCL-X$_L$ by PROTACs reduces the frequency of TI-Tregs by inducing apoptosis, resulting in the activation of TI-CD8[+] T cells. This leads to the inhibition of tumor growth without significant induction of serum immunoglobulins and/or detectable tissue damage from lung, intestine, and pancreas. Interestingly, a recent report from Dr. Zou group found that the apoptotic Tregs were able to induce a strong immune suppression via the CD39/CD73-mediated conversion of ATP to immunosuppressive metabolite adenosine in human ovarian cancers[26]. In the current study, we determined the kinetics of DT2216-induced caspase activation within TI-Tregs and found that caspase activation was detectable at 24 h after DT2216 treatment of MC38 tumor-bearing mice, but completely disappeared after 48 h of treatment (Supplementary Fig. 10a, b), supporting the rapid clearance of these apoptotic TI-Tregs within tumor microenvironment. Apoptotic clearance by monocytes/macrophages is believed to be an event starting from earlier stages of apoptotic induction[28,29]. Ly6C[+] monocytes or tumor-associated macrophages (TAMs) are much more abundant populations in most cancer models and human cancers than TI-Tregs. The battle between the clearance of apoptotic Tregs and the maintenance of Treg-suppressive functions—in our case—clearly favors the clearance of apoptotic TI-Tregs.

As shown in the immune checkpoint inhibitor therapies, it is likely that BCL-X$_L$ PROTAC-based immunotherapy will be effective in a subset of cancer patients. It will be important to identify the relevant populations, as well as cancer types that are susceptible to this treatment, before clinical trials with BCL-X$_L$ PROTACs are considered. Cancer types that are likely to be most susceptible to treatment with DT2216 include: (1) those in which TI-Tregs are critical for immune regulation and cancer progression; and (2) those that rely on BCL-X$_L$ for survival and in which high BCL-X$_L$ levels and poor outcome are positively correlated. Cancer types that rely on TI-Tregs for their progression include several common ones, for example cervical, renal, ovarian, melanoma, pancreatic, gastric, and breast cancers. In these cancer types, DT2216 may be able to eliminate TI-Tregs and thereby boost immunity against cancers[30]. To identify cancer types that rely on BCL-X$_L$ for survival, we used our recently developed cloud-based application[31] to perform a bioinformatics-based search of the Cancer Genome Atlas (TCGA) protein dataset. We found that BCL-X$_L$ levels correlated significantly with hazard ratios (HR) within renal, melanoma, breast, and cervical cancers, as well as sarcomas and low-grade gliomas. These are cancers for which DT2216 treatment might lead to direct killing of cancer cells and/or sensitization to anti-tumor immunity (Supplementary Fig. 11). The expression of VHL and CRBN suggests that the choice of E3 ligase will also be critical in selecting BCL-X$_L$ PROTACs. For example, VHL is frequently lost in certain cancer types, such as RCC; in such cases, a different E3 ligase such as CRBN would be more appropriate. This study provides the rationale and a proof-of-principle to eliminate Tregs within tumors by targeting and degrading an intracellular pro-survival factor BCL-X$_L$. BCL-X$_L$ PROTACs may provide additional choice for cancer immunotherapies particularly within cancer patient populations that cannot be treated by the current cancer immunotherapies due to low efficacy or immune toxicity.

## Methods

**Human tissue collection, isolation of mononuclear cells, and sorting for scRNA-seq**. Paired blood and primary ccRCC along with matched normal kidney parenchyma samples were obtained from the University of Iowa Tissue Procurement Core and GUMER repository through the Holden Comprehensive Cancer Center from de-identified three subjects (with an age range of 67–74 years old) previously provided written consent approved by the University of Iowa Institutional Review Board (IRB) under the IRB number 201304826 and conducted under the Declaration of Helsinki Principles. Tumor grades were histologically determined by a pathologist. Primary tumor stages for Patient 1 and Patient 2 were reported as pT1b without extension, while Patient 3 was reported as pT3a with renal vein invasion. Other human tissues were collected under IRB protocol: 201901677 that was approved by the University of Florida Institutional Review Board as non-human subject protocol, including all tissues for ex vivo culture of tumor slice sections. Tumor samples were dissociated into single cells via enzymatic and mechanical dissociation using the Human Tumor Dissociation Kit (Miltenyi Biotec, Auburn, CA). CD45[+] cells were isolated from single-cell suspensions using the SepMate FITC Positive Selection Kit (StemCell Technologies, Vancouver, Canada). Mononuclear cells were isolated from PB samples by density gradient using SepMate Tubes (StemCell Technologies) and Lymphoprep density gradient media (StemCell Technologies). CD45[+] immune cells were further sorted on a FACS ARIA sorter (BD Biosciences, San Jose, CA) for lymphoid and myeloid cells and remixed at a 3:1 ratio. Cell viability was assessed by MoxiGoII counter (Orflo Technologies, Ketchum, ID) and resuspended at 1000 cells/μl with >90% viability.

**RNA-seq 10X Genomics library preparation, sequencing, and alignment**. Single-cell library preparation was carried out as per the 10X Genomics Chromium 5′ library and Gel Bead Kit Version 2 (10X Genomics, Pleasanton, CA). Following cDNA amplification, quality control and quantification were performed on the Agilent 2100 Bioanalyzer using the DNA high Sensitivity Chip (Agilent Technologies, Santa Clara, CA). Pooled Libraries were run on separate lanes of a flow cell using the Illumina HiSeq 4000 in the University of Iowa Genomics Division. Basecalls were converted to FASTQ files using the Illumina bcl2fastq software and aligned to human genome (GRCh38) using the CellRanger v2.2 pipeline as described previously[32]. Cells were quality checked for total expression of mitochondrial reads. Cells with <200 or >5000 unique genes were filtered out.

**Single-cell RNA normalization and analysis**. The single-cell RNAseq data were normalized by scaling the unique molecular identifier (UMI) counts per cell to 10,000 and the data were Napierian $\log(n + 0.1)$ transformed. Clustering was performed using the Seurat R package (V2.3.4) and the data were normalized across the three patients to correct for patient variability using the canonical correlation process[33]. Dimensional reduction using the top 20 calculated dimensions and resolution of 1.2 was done to generate the tSNE plot. The predicted cell type for each cluster was determined by examining the expression levels of a set of canonical markers for each cell type and by correlating the cluster mean expression to immune cell signatures from human primary cell atlas dataset using the SingleR R package V0.2.0 (ref. [34]). A fold change in the expression of BCL-2 family genes between TI T cells and PB-T cells was determined by comparing the average Napierian $\log(n + 0.1)$-transformed counts for BCL2 family member genes for each of the four clusters of TI cells identified as CD8[+] T cells to the single-cluster PB cells identified as CD8[+] T cells, the average Napierian $\log(n + 0.1)$-transformed counts of the single cluster of TI cells identified as CD4[+] T cells to each of the three clusters of PB cells identified as CD4 T cells, and TI cells to the PB cells in the cluster identified as Tregs. The fold change values were used to generate a heat map using GraphPad Prism Software (GraphPad Software, San Diego, CA). scRNAseq data for human HCC data from GSE98638 (ref. [19]) underwent the same processing and quality control procedures. The average Napierian $\log(n + 0.1)$ expression of BCL2 family genes in all TI CD8[+], TI CD4[+], and TI Tregs were compared to their PB counterparts and a heatmap of the Napierian log-fold change for each gene was made using GraphPad Prism Software. Violin plots of the expression of genes in TI and PB Tregs were generated using monocle 2R Package.

**Bulk RNA sequencing and analysis**. Mouse MC38 tumors were enzymatically and mechanically dissociated to form single-cell suspensions using the Mouse Tumor Dissociation Kit (Miltenyi Biotec) and mononuclear cells were isolated by density gradient using SepMate Tubes and Lymphoprep (StemCell Technologies). Single-cell suspension of splenic cells was obtained by pressing spleens from the same tumor bearing mice through a 40-μm nylon strainer and lysing red blood cells using ACK lysis buffer (150 mM $NH_4Cl$, 10 mM $KHCO_3$, 0.1 mM EDTA). Cells were then stained with mouse anti CD45-FITC (clone 30F.11; Biolegend), CD3-APC (clone 17A2, Biolegend), CD4 (clone GK1.5; Biolegend), CD25-PE-Cy5 (clone PC61, Biolegend), and glucocorticoid-induced TNFR-related (GITR, clone DTA1; Biolegend) and Fixable, Viability Dye (FVD)-eFluor (EF) 780 (eBioscience from Thermo Fisher Scientific, San Diego, CA). All antibodies were used here at 1:100 dilution for flow cytometry. Live Tregs (CD4[+]GITR[+]CD25[+]) were sorted using a BD LSRFortessa SORP (BD Biosciences, San Jose, CA), into RLT buffer and RNA was isolated using the RNeasy Plus Micro Kit (Qiagen, Hilden, Germany). Ribosomal RNA was depleted using the NEBNext Ultra II rRNA depletion Kit (New England Biolabs, Ipswich, MA) and libraries were generated using the NEBNext Ultra II RNA Library Prep Kit (New England Biolabs). Libraries were sequenced on a NextSeq 500 $1 \times 75$ base pairs at an approximate read depth of 30 million reads per library. Samples were aligned with the kallisto pseudoalignment protocol and mm10 build for the mouse genome to produce estimated counts[35]. Counts were then processed using the sleuth R package (v0.30.0)[36] aggregating based on gene symbol. Expression values were transformed into $\log2(n + 1)$ values.

**Chemical synthesis**. DT2216 (ref. [16]) and PZ15227 (ref. [18]) were synthesized as described previously.

**Ex vivo culture of tumor slice sections**. De-identified human tumor samples were obtained from the Clinical and Translational Science Institute Biorepository at the University of Florida under the IRB approved study number IRB201901677. Fresh human or mouse tumor samples were encased in 3% agarose and cut into 300-μm-thick sections using a Compresstome VF-300 OZ (Precision Instruments, Greenville, NC). Sections were cultured in 12-well plates in Dulbecco's Modified Eagle Medium (DMEM) supplemented with 10% fetal bovine serum (FBS), 100 U/ml penicillin, and 100 μg/ml streptomycin. Alternating slices were grouped and treated with different concentrations of DT2216 (25 nM, 50 nM, and/or 100 nM) or vehicle alone and cultured at 37 °C and 5% $CO_2$ for 48 h. Tumor slice samples were dissociated into single cells via enzymatic and mechanical dissociation using the Human Tumor Dissociation Kit (Miltenyi Biotec), following with flow cytometry to quantitate different populations of immune cells.

**Cell lines and cell culture**. MC38 mouse colon carcinoma cells (Kerafast Inc., Boston, MA) were maintained in DMEM supplemented with 10% FBS, 1 mM glutamine, 0.1 M non-essential amino acids (Thermo Fisher, Waltham, MA), 1 mM sodium pyruvate, 10 mM HEPES, 100 U/ml penicillin, and 100 μg/ml streptomycin. 4T1 mouse mammary carcinoma cells and Renca mouse renal carcinoma cells were purchased from American Type Culture Collection (ATCC) and cultured in RPMI medium supplemented with 10% FBS, 100 U/ml penicillin, and 100 μg/ml streptomycin. Py8119 mouse mammary carcinoma cells[37] were obtained from Dr. Lesley Ellies at the University of California San Diego and cultured in DMEM supplemented with 10% FBS, 100 U/ml penicillin, and 100 μg/ml streptomycin. All cells were cultured at 37 °C and 5% $CO_2$ in a humidified incubator.

**CRISPR/CAS9-mediated KO and viability assay**. BCL-$X_L$-KO MC38 and Renca cells were generated by transient transfection of Cas9 protein-guide RNA complex into parental cells, followed by single clone section and immunoblot to confirm KO of BCL-$X_L$ following the instruction (Synthego). The guide RNA used for BCL-$X_L$ KO is 5′-AUACUUUUGUGGAACUCUAU-3′.

Cells were plated in 96-well plates at a density of $3 \times 10^3$ to $5 \times 10^3$ cells per well and cultured overnight. Cells were treated with increasing concentrations of DT2216, PZ15227, ABT199, a BCL2 specific inhibitor[10] (LC laboratories), A1155463, a BCL-$X_L$ specific inhibitor[38] (Selleck Chemicals, Houston, TX), ABT263, a BCL-2/BCL-$X_L$ dual inhibitor[9] (Selleck Chemicals) or S63845, a MCL-1 selective inhibitor[12] (Selleck Chemicals) for 72 h. Cell viability was measured by tetrazolium-based MTS assay as described previously[16]. Briefly, MTS reagent (Promega, Madison, WI) was supplemented with phenazine methosulfate (Sigma-Aldrich, St. Louis, MO) at a 20:1 ratio and 20 μl was added to each well and incubated for 4 h at 37 °C. Absorbance was measured at 490 nm using a Synergy Neo2 multimode plate reader (Biotek, Winooski, VT) and cell viability was determined for each well. The data were expressed as the average percent of viable cells and fitted in non-linear regression curves using Prism software (GraphPad Software).

**Animal studies**. All animal studies were approved by the University of Florida Institutional Animal Care and Use Committee (IACUC) under protocol 201810399. All animals were housed in a pathogen-free Association for Assessment and Accreditation of Laboratory Animal Care accredited facility at the University of Florida and performed in accordance with IACUC guidelines. The room temperature is 21.1–23.3 °C with an acceptable daily fluctuation of 2 °C. Typically the room is 22.2 °C all the time. The humidity set point is 50% but can vary ±15% daily depending on the weather. The photoperiod is 12:12 and the light intensity range is 15.6–25.8 FC. C57BL/6 and Balb/C mice were purchased from Charles River Laboratories (San Diego, CA). NOD-SCID interleukin-2 receptor gamma null (NSG) mice were purchased from Jackson Laboratories (Bar Harbor, ME). For all studies involving DT2216 or PZ15227 (ref. [16]), the compounds were formulated in 50% phosal PG, 45% miglyol 810N, and 5% polysorbate 80 and administered via intraperitoneal (i.p.) injection. ABT199 (LC Laboratories, Woburn, MA) was formulated in 60% phosal 50 PG, 30% PEG 400, and 10% ethanol and administered 5 consecutive days per week by oral gavage. For studies in non-tumor bearing mice, 6–8-week-old C57BL/6 mice were treated once weekly with DT2216 at 7.5 mg/kg body weight or vehicle alone. Blood samples were taken just before each treatment and 24 h later. For syngeneic tumor models, $1 \times 10^5$ cells (MC38 and Renca) or 500 cells (for Py8119) were resuspended in 50% phosphate buffered saline (PBS), 50% Matrigel (Corning Inc., Corning, NY), and implanted into 6–8-week-old C57BL/6 (MC38, s.c.; and Py8119, intra mammary fatpad), Balb/C (Renca, s.c.), or NSG mice. Tumor growth was monitored daily and measured twice weekly with calipers and volume was determined using the formula ½ $(L \times W^2)$. Once the largest tumors reached 0.5 cm in diameter, mice were treated with 7.5 mg/kg body weight DT2216 by i.p. injection once weekly, or with 30 mg/kg body weight of PZ15227 by i.p. injection every 4 days. For BCL-2 inhibitor study, once the largest tumors reached 0.5 cm in diameter, mice were treated with 100 mg/kg ABT199 5 days per week by oral gavage. Mice were euthanized in accordance with IACUC protocol once the largest tumors reached 2 cm in diameter and tissues were collected for further analysis.

For non-tumor bearing mice, 8-week-old mice were treated with 7.5 mg/kg body weight DT2216 by i.p. injection once weekly for upto 5 weeks when tissues were collected for analysis.

**Flow cytometry**. Mouse tumors were excised and ~200 mg of tumor tissue was enzymatically and mechanically digested using the mouse Tumor Dissociation Kit (Miltenyi Biotec) to obtain a single-cell suspension. Human tumor samples and sections were enzymatically and mechanically digested using the human Tumor Dissociation Kit (Miltenyi Biotec) to obtain single-cell suspension. Red blood cells were lysed using ACK lysis buffer and mononuclear cells were isolated by density gradient using SepMate Tubes (StemCell Technologies) and Lymphoprep density gradient media (StemCell Technologies). Mouse cells were then washed and incubated with combinations of the following antibodies; anti-mouse CD62L-BV785 (clone MEL-14), anti-mouse MHCII I-A/I-E-BB515 (Clone 2G9, BD Biosciences, 1:400), anti-mouse CD11B-PEdazzle (clone M1/70, 1:200), anti-mouse CD45-AF532 (clone 30F.11), anti-mouse CD3-APC (clone 17A2), anti-mouse CD8-BV510 (clone 53-6.7), anti-mouse CD4-BV605 (clone GK1.5), anti-mouse NK1.1-AF700 (clone PK136), anti-mouse CD69-SB436 (clone H1.2F3, eBioscience), anti-mouse CD279 (clone-PerCP-EF710, eBioscience Inc), anti-mouse CD366-PacBlue (clone B8.2c12), anti-mouse CD11C-PE-Cy7 (clone N418), anti-mouse Ly6G-FITC (clone IA8), anti-mouse Ly6C-BV711 (clone HK1.4), anti-mouse F4/80-BV650 (clone BM8), anti-mouse CD80-BV480 (clone 16-10A1, BD Biosciences), anti-mouse CD25-PE-Cy5 (clone PC61) plus FVD-eFluor-780 (eBioscience) and mouse FcR blocker (anti-mouse CD16/CD32, clone 2.4G2, BD Biosciences). After surface staining, cells were fixed and permeabilized using the FOXP3/Transcription Factor Staining Buffer Set (eBioscience). Cells were stained with a combination of the following antibodies: anti-mouse FOXP3-APC (clone FJK-16S, 1:50, eBioscience), anti-mouse Granzyme B-Pacific Blue (clone GB11), anti-mouse Perforin-PE (clone S16009B), anti-mouse Ki-67-PerCP-Cy5.5 (clone 16A8), and anti-mouse/human BCL-$X_L$-PE (clone S486, 1:50, Cell Signaling Technologies, Danvers, MA). Human cells were stained with a combination of the following

antibodies: anti-human CD45-BV510 (clone H130), anti-human CD3-AF700 (clone HIT5a), anti-human CD4-BV421 (clone OKT4), anti-human CD8-BV711 (clone RPA-T8), anti-human CD127-BV605 (clone A019DS), anti-human CD25-PE-Cy7 (clone MA251) plus FVD-eFluor-780 (eBioscience) and human FcR blocking Reagent (StemCell Technologies). Anti-human CD45RA (clone: HI100) and anti-human CCR7 (clone: G043H7) were included for the effector/memory analysis for different populations of PB and TI T cells. Cells were washed then fixed and permeabilized using the eBioscience FOXP3/Transcription Factor Staining Buffer Set. Cells were further stained with a combination of the following antibodies: anti-human FOXP3-FITC (clone 206D), anti-mouse/human BCL-X$_L$-PE (clone S486, Cell Signaling Technologies). Flow cytometry was performed on a 3 laser Cytek Aurora Cytometer (Cytek Biosciences, Fremont, CA) and analyzed using FlowJo software (BD Biosciences). All antibodies are from Biolegend, unless otherwise specified. Most antibodies were used at 1:100 dilution for flow cytometry, unless specified.

**NIR-FLIVO poly-caspase activity assay**. MC38 tumor-bearing mice were treated with 7.5 mg/kg DT2216 or vehicle via i.p. injection; 24 h later, the mice were injected with 50 µl of NIR-FLIVO 690 caspase probe (ImmunoChemistry Technologies, Bloomington, MN) via i.v. injection per the manufacturer's suggestion. Tumors and spleens were harvested after 4 h. Tumors were chemically and enzymatically digested using the mouse Tumor Dissociation Kit (Miltenyi Biotec) and mononuclear cells were isolated by density gradient using SepMate 50 tubes (StemCell Technologies) and lymphoprep (StemCell Technologies). Spleens were pressed though a 40-µm nylon strainer and red blood cells were lysed with ACK lysis buffer. Cells were incubated with the following antibodies: anti-mouse CD45-AF532 (clone 30F.11), anti-mouse CD3-APC (clone 17A2), anti-mouse CD4-BV605 (clone GK1.5), anti-mouse CD8 (clone MEL-14), CD11B-PE-dazzle (clone M1/70, 1:200), and FVD-eFluor-780. Cells were then fixed and permeabilized with eBioscience FOXP3/Transcription Factor Staining Buffer Set, following with intracellular staining of anti-mouse FOXP3-EF450 (FJK-16S, eBioscience, 1:50). Antibodies were used at 1:100 dilution for flow cytometry unless specified. Flow cytometry was performed on a 3 laser Cytek Aurora Cytometer (Cytek Biosciences, Fremont, CA) and analyzed using FlowJo software (BD Biosciences). All antibodies are from Biolegend, unless otherwise specified.

**Immunoblot**. Cells or tumors samples were lysed with RIPA buffer (150 mM NaCl, 5 mM EDTA, 50 mM Tris pH 8.0, 1% sodium deoxycholate, 1% NP-40, 0.5% SDS) supplemented with 1 mM dithiothreitol and protease inhibitors. Cell lysates were then separated by SDS-PAGE and analyzed by standard western blotting protocol. Antibodies used were anti-BCL-X$_L$ (Cell Signaling, 54H6, 1:1000), anti-MCL-1 (Cell Signaling, D35A5, 1:1000), anti-BCL-2 (Cell Signaling, 50E3, 1:500), and anti-β-actin (Cell Signaling, 1:10,000).

**Statistical methods**. All graphs, unless indicated otherwise, were made and statistical analysis done using Prism software (GraphPad Software). For comparison of tumor growth curves, a two-way analysis of variance (ANOVA) was performed. To compare the means of three or more groups, a one-way ANOVA was performed and comparisons between specific groups were done using Dunnett's multiple comparison test. To compare the means of two groups, an unpaired $T$ test was performed. For analysis of RNAseq expression data comparing the expression of genes between two groups, a two-tailed unpaired $T$ test was performed with a Bonferroni correction. The precise $P$ values were provided.

**Reporting summary**. Further information on research design is available in the Nature Research Reporting Summary linked to this article.

## Data availability
We included analysis from publicly available datasets including: GSE89225 (ref. [21]) and GSE98638 (ref. [19]). The single-cell RNA sequencing result for renal cancer is deposited as GSE121638 and for the TI-Tregs from MC38 tumor model, it is deposited as GSE150420. The remaining data are available within the Article, Supplementary Information, or available from the authors upon request.

## Code availability
All modified codes related to SCRC analysis are deposited at: https://github.com/ncborcherding/BCLXL.

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

## Acknowledgements
This study was supported in part by NIH grants CA200673 (W.Z.), CA203834 (W.Z.), CA206255 (N.B.), CA242003 (G.Z. and D.Z), and CA219836 (D.Z.), DOD/CDMRP grant BC180227 (W.Z.), and an endowment from the Dr. and Mrs. James Robert Spencer Family Cancer Research Fund (W.Z.). We thank Dr. Haoyang Zhuang and Ms. Mingjia Li from University of Florida for their help in autoantibody measurement.

## Author contributions
R.K. and U.D. performed and analyzed most of the experiments and wrote the manuscript; Xuan, Z., P.Z., and G.Z. designed, synthesized, and formulated the BCL-XL PROTACs. S.K. designed and performed the in vitro drug sensitivity assays. M.K., Y.L., H.Y., J.M., D.K., and C.W. performed the in vitro experiments with cancer cell lines. N.B. performed the bioinformatics analysis of the scRNAseq dataset; Xin, Z. performed some flow cytometry. D.A., Y.X.F., and S.G.Z. contributed to critical discussions, concept development, and manuscript editing. W.Z., D.Z., and R.K. conceived, designed, and supervised the study, analyzed and interpreted the data, and wrote the manuscript. All authors discussed the results and commented on the manuscript.

## Competing interests
S.K., Xuan, Z., G.Z., and D.Z. are inventors of two pending patent applications for the use of BCL-XL PROTACs as senolytic and antitumor agents. G.Z. and D.Z. are co-founders of, and have equity in, Dialectic Therapeutics, which develops BCL-XL PROTACs for the treatment of cancer. The other authors have no competing interests.
