## [Peer Review File · Nature Communications]

REVIEWER COMMENTS

Reviewer #1 (Remarks to the Author): with expertise in BCL2 protein targeting

In this study, Kolb et al investigated the role of BCL2 proteins in protecting TI-Tregs in from the pro-apoptotic immunosuppressive tumour microenvironment. They performed single cell RNA-Sequencing of TI-Tregs in comparison to PB lymphocytes and Tregs derived from the same patients. By using both their own data as well as published datasets they discovered a selective upregulation of BCL-XL in TI-Tregs compared to PB-Tregs and other TI-lymphocytes across different cancer entities.

To target BCL-XL they previously developed DT2216, a BCL-XL-PROTAC, which links BCL-XL to the VHL E3 ligase and hence does not show platelet toxicity. This compound decreases BCL-XL in the TI-Tregs and shows antitumour activity in vivo without having a direct effect on tumour cell viability in the models used here.

Overall, the data presented are convincing and this study is of great interest to advance the translation of BCL-XL-targeting agents into effective anticancer treatments.

Minor comments:

1. In Fig 4f the labelling says XZ15807 but DT2216 is mentioned in the legend.
2. Open bracket on P4 after "(HCC scRNAseq dataset GSE98638"
3. The statement on P5 sounds misleading: "To assess the effectiveness of PROTAC-mediated inhibition of BCL-XL degradation..." Does this compound not induce BCL-XL degradation, rather than inhibit it?
4. ExtData Fig 7a: Please provide a legend. Is this a comparison of tumour growth in wt versus BCL-XL KO without any treatment? Which of the clones depicted in the inset was used in these experiments?

Signed by Meike Vogler

Reviewer #2 (Remarks to the Author): with expertise in Treg

The manuscript from Kolb et al., "Proteolysis-targeting chimera against BCL-xL destroys tumor-infiltrating regulatory T cells", addresses an important hypothesis; that targeting BCL-xL can specifically kill suppressive T cells and enhance immunity to cancer without immune side effects. The authors use innovative approaches and a recently published PROTAC targeting BCL-xL to test this concept. Overall, the study is interesting and novel; however, there are a range of issues with the data and their interpretation that, in my view, currently make the manuscript unsuitable for publication in Nature Communication. The major issues are: 1) demonstration of the mechanism by which the PROTAC impacts on tumour growth, 2) the claim that the approach does not induce autoimmunity and 3) the stringency of the comparisons made to establish that the upregulation of BCL-xL in Treg cells is specific to the tumour.

1) Figure 4 shows that targeting BCL-xL with the PROTAC reduces (slows?) tumour growth. While the data in immunodeficient mice establish a role for the immune system in this process, the authors do not characterise the tumours in immunocompetent recipients at all. A central point of the paper is that the mechanism by which the PROTACs work is by killing Treg cells, so they really need to show (at least) that Treg cells are reduced/apoptotic in these tumours and that conventional T cell activation increased. There are many other suppressive immune cells (e.g. myeloid suppressor cells, Tr1 regulatory cells, Bregs) that could be involved in this effect. There are also less direct explanations, such as enhanced antigen presentation, that could account for the effects observed. A setting where Treg cells specifically could not be targeted by the PROTAC would provide the ideal negative control and an excellent test of the author's hypothesis.

2) A major strength of the BCL-xL PROTAC is that it avoids the thrombocytopenia induced by pharmacological BCL-xL inhibitors. However, the authors data show that it depletes Treg cells in

lymphoid tissues; not just the tumours (Fig 3A). Aside from a perfunctory analysis of three organs, there is no assessment of autoimmunity in the mice treated with the BCL-xL-targeting PROTAC. Is there increased T cell activation? Are there higher circulating autoantibodies in treated mice? What about other tissues? The authors need these data to support their key claim that there are no autoimmune side-effects with this treatment (Abstract, Discussion).

3) The conclusion that Treg cells infiltrating tumours specifically upregulate BCL-xL, and that other immune cell types do not, is tested broadly, but not very deeply. In most cases, the comparison made is between Treg cells infiltrating the tumour and conventional CD4+ and CD8+ T cells subsets in lymphoid tissues or the blood. CD4+ and CD8+ T cell populations are composed of many functionally distinct subtypes (e.g. naïve, activated, effector, memory) and their representation in lymphoid recirculation and tissues is very different (e.g. many more naïve T cells). Since the expression profiles of BCL-2 family proteins also varies greatly among these subtypes, a more suitable comparison would involve tumour Treg vs. effector/memory CD4+ and CD8+ T cells from lymphoid tissues and non-lymphoid tissues. There is also heterogeneity among Treg cell populations that should also be taken into account. Are all Treg subtypes in tumours sensitive to the PROTAC? Moreover, only 3 patients per tumour type were assayed. Given normal human variation (and among tumours), I am very surprised such tiny standard deviations are reported in Figures 1D, E, G and H. There is also no information on the tumour types or their staging, which makes it harder to discern how broadly this observation applies and how others might reproduce the findings.

Minor comments:

1) The data from the FLIVO experiments are not convincing. These data are important because they are the only experiments testing whether apoptosis is the mechanism of Treg cell reduction. The FLIVO shift in Ext. Fig. 4B is modest (at best) and there are no controls guiding the placement of the gate/marker used.

2) The in vitro cancer slice experiments are innovative, but I am concerned that there would be a lot of basal apoptosis in certain cell types due to hypoxia. This issue might influence the outcomes observed based on the localisation of immune cell types. Further data characterizing this system would be helpful. How long between harvesting the tissue and establishment of the cultures? Also, when samples sizes of 3-4 are quoted, do these refer to the number of slices from an individual donor, or the number of donors assayed?

3) There are no statistical tests applied to the RNAseq data; only expression fold-changes are reported. Some indication of how reproducible or robust these observations are is essential.

4) The flow cytometry of BCL-xL expression is important and a strong element in the paper. It would be better if unstained controls were added so readers could assess the extent of the change in MFI and whether the low signals (say, in PB-Treg in Fig 1C) represents low or negative staining. This is an important issue when interpreting potential side-effects of the PROTAC-targeting of BCL-xL.

Reviewer #3 (Remarks to the Author): with expertise in Treg and tumor immunology

This manuscript describes the potential application of BCL-XL Proteolysis-Targeting Chimera (PROTAC) as a regulatory T (Treg) cell-targeted reagent. Treg cells play an important role in tumor immunity by suppressing a wide range of antitumor immune responses. Indeed, high frequency of Treg cells and their dominance to effector T cells in the tumor microenvironment (TME) are associated with poor prognosis in various types of cancer. Yet, Treg cell-targeted therapies have not been applied into the clinic, and developing strategies to control Treg cells is an urgent issue in the cancer immunotherapy field. The authors addressed the issue with focusing on high and selective BCL-XL expression by tumor infiltrating Treg cells. BCL-XL was highly expressed by tumor infiltrating Treg cells compared to peripheral Treg cells and other effector T cells such as CD8+ T cells and CD4+ non-Treg cells in human cancers and murine tumor models. Accordingly, DT2216, the lead BCL-XL PROTAC induced selective reduction of tumor infiltrating Treg cells via apoptosis

induction, resulting in the activation of effector T cells. While one major concern with targeting BCL-XL is concomitant reduction of platelets, platelets were not influenced as their DT2215 employed VHL E3 ligase for polyubiquitination and degradation. Furthermore, DT2216 exhibited a strong antitumor efficacy in various tumor models, and the antitumor effects were dependent on CD8+ T cells.

General comments;

Given a crucial role of Treg cells in self-tolerance, Treg cell depletion as a whole can trigger autoimmunity in animal models, thereby how Treg cells can be controlled to evoke and augment antitumor immunity without inducing deleterious autoimmunity, strongly indicating the necessity of developing Treg-cell depletion methods with high selectivity to tumor-infiltrating Treg cells. This paper documents a very interesting approach with their novel Treg cell-targeted reagent. Targeting BCL-XL for selective Treg cell deletion is a novel concept and their preclinical study data are very promising. Yet, lacking some experiments needs to be revised because of missing important controls (see Specific comments).

Specific points;

1. In Figure 2 and 3, the authors show the selective reduction of Treg cells by apoptosis induction. However, there is a contradictory paper showing that adenosine produced by apoptotic Treg cells induces a strong immune suppression, and macrophage-associated clearance plays an important role for effective immune activation via Treg cell reduction by apoptosis (Maj et al Nat Immunol 18 1332-1341 2017). The authors should address the potential immune suppression by Treg cell apoptosis in their system.
2. In Figure 3 and Extended Figure 4, the effects of DT2216 on other immune cells are shown. A previous human Treg cell-targeted treatment with anti-CCR4 mAb revealed the complementary expansion of other immune suppressive cells such as MDSCs upon Treg cell depletion (Kurose et al Clin Cancer Res 21 4327-4336 2015). The authors need to show the kinetics of other immune suppressive cells.
3. In Figure 3 and Extended Figure 7, expansion of entire CD8+ T cells upon Treg cell depletion is shown. The authors need to examine tumor antigen-specific CD8+ T cell responses.
4. DT2216 exhibited a strong antitumor efficacy in various tumor models by depleting Treg cells. Presumably there would be no synergy with other Treg cell depletion methods (anti-CTLA-4 mAb / DTR treatment in FoxP3-DTR mice). By contrast, how about the synergistic effect with PD-1 blockade treatment.
5. There are missing control staining in histograms in Figure 1-3 and Extended Figure 2 and 4.
6. The references do not appear in order, e.g. 1 followed by 4.

Reviewer #4 (Remarks to the Author): with expertise in scRNA seq analysis

The manuscript "Proteolysis-targeting chimera against BCL-XL destroys tumor-infiltrating regulatory T cells" from Kolb, De and co-authors describe the impact of pharmacological blockage of BCL-XL with a Proteolysis-Targeting Chimera in tumor cell growth. BCL-xl blocking leads to Treg depletion in the tumor and enhanced CD8 T cell response, which leads to reduced tumor growth.

The manuscript makes use of a range of different techniques and models to identify BCL-xl expression in tumor infiltrating lymphocytes and the effects of its pharmacological targeting. The authors analyse a large number of different human and mouse tumor types, which is a very positive aspect of the work, as it offers potential in a broad range of cancer types. This is an

impressive and positive aspect of the work, and I have just a few minor comments below.

1) To improve readability, due to the large number of different cell lines and cancer models used, I suggest adding in each figure legends the cancer type of the cells at least once. This is present in figures 1 and 2, but not 3 and 4 and many of the supplementary cells.

2) The analysis of the single cell rna-seq data generated by the authors is rather superficial, which reduces the re-usability of the dataset. I advise for inclusion of more detailed panels, indicating proper gene markers and better cluster assignments in extended figure 1.

3) also on supplementary figure 1, the depiction of % difference versus log-fold change would benefit from a proper statistical measure of difference (p-value).

4) In the data processing methods parts, the authors write "The data were then $\log(n+1)$ transformed" and a few sentences later in the same page, "the average \log_2 transformed". The correct log conversion should be stated.

Reviewer 1:

In this study, Kolb et al investigated the role of BCL2 proteins in protecting TI-Tregs in from the pro-apoptotic immunosuppressive tumour microenvironment. They performed single cell RNA-Sequencing of TI-TRegs in comparison to PB lymphocytes and TRegs derived from the same patients. By using both their own data as well as published datasets they discovered a selective upregulation of BCL-XL in TI-TRegs compared to PB-TRegs and other TI-lymphocytes across different cancer entities.

To target BCL-XL they previously developed DT2216, a BCL-XL-PROTAC, which links BCL-XL to the VHL E3 ligase and hence does not show platelet toxicity. This compound decreases BCL-XL in the TI-TRegs and shows antitumour activity in vivo without having a direct effect on tumour cell viability in the models used here.

Overall, the data presented are convincing and this study is of great interest to advance the translation of BCL-XL-targeting agents into effective anticancer treatments.

Response: We thank the reviewer for his/her strong support of our manuscript and their appreciation of the value of translating BCL-XL PROTACs into the clinic. In fact, DT2216 is currently under advanced preclinical development right now. We hope that it can be translated into the clinic early next year after receiving the IND approval from the FDA.

Minor comments:

1. *In Fig 4f the labelling says XZ15807 but DT2216 is mentioned in the legend.*

Response: XZ15807 was the previous name for DT2216. We changed it to DT2216 as previously published in our Nature Medicine publication (1) (new Fig.3f).

2. *Open bracket on P4 after "(HCC scRNAseq dataset GSE98638"*

Response: Thanks and corrected.

3. *The statement on P5 sounds misleading: "To assess the effectiveness of PROTAC-mediated inhibition of BCL-XL degradation..." Does this compound not induce BCL-XL degradation, rather than inhibit it?*

Response: We are sorry for the mistake. This compound induces BCL-XL degradation, rather than inhibit it. It has been corrected to "To assess the effectiveness of PROTAC-mediated BCL-XL degradation...".

4. *ExtData Fig 7a: Please provide a legend. Is this a comparison of tumour growth in wt versus BCL-XL KO without any treatment? Which of the clones depicted in the inset was used in these experiments?*

Response: We thank the reviewer for the suggestion. A figure Legend is added. WT cells (black line) and

clone 2 BCL-X_L knockout cells (red line) were used in the study without any treatment (new **Extended Data Fig. 6a**).

Reviewer #2 (Remarks to the Author): with expertise in Treg

The manuscript from Kolb et al., “Proteolysis-targeting chimera against BCL-xL destroys tumor-infiltrating regulatory T cells”, addresses an important hypothesis; that targeting BCL-xL can specifically kill suppressive T cells and enhance immunity to cancer without immune side effects. The authors use innovative approaches and a recently published PROTAC targeting BCL-xL to test this concept. Overall, the study is interesting and novel; however, there are a range of issues with the data and their interpretation that, in my view, currently make the manuscript unsuitable for publication in Nature Communication. The major issues are: 1) demonstration of the mechanism by which the PROTAC impacts on tumour growth, 2) the claim that the approach does not induce autoimmunity and 3) the stringency of the comparisons made to establish that the upregulation of BCL-xL in Treg cells is specific to the tumour.

Response: We would like to thank the reviewer for his/her insightful suggestions. His/her main concerns have been addressed below:

1) Figure 4 shows that targeting BCL-xL with the PROTAC reduces (slows?) tumour growth. While the data in immunodeficient mice establish a role for the immune system in this process, the authors do not characterise the tumours in immunocompetent recipients at all. A central point of the paper is that the mechanism by which the PROTACs work is by killing Treg cells, so they really need to show (at least) that Treg cells are reduced/apoptotic in these tumours and that conventional T cell activation increased.

Response: We appreciate the reviewer’s suggestion. We carefully evaluated apoptosis/viable cells and changes in the numbers and percentages of Treg, Tconv (CD4 conventional T cells), CD8 T cells and CD8 T cell activation in the tumor microenvironment after DT2216 treatment as suggested by the reviewer. In addition, we profiled the changes in all other myeloid and lymphoid cells in the tumor microenvironment as well. The results from these analyses are presented in the new **Fig. 4** and **Extended Data Fig. 7-9** in the revised manuscript (please note that we switched the original *Fig. 3* and *Fig. 4*, as well as relevant extended data figures to make the manuscript flow better per reviewer’s suggestion). The data presented in these figures demonstrate that DT-2216 selectively depletes TI-Tregs and activates TI-CD8 T cells, while having no significant effect on other cell populations (please see our response below).

There are many other suppressive immune cells (e.g. myeloid suppressor cells, Tr1 regulatory cells, Bregs) that could be involved in this effect. There are also less direct explanations, such as enhanced antigen presentation, that could account for the effects observed.

Response: Great suggestions. We thoroughly analyzed the major myeloid cell populations in the tumors as suggested. We did not see significant differences in the number of monocytic myeloid derived suppressor cells (M-MDSCs; CD11b⁺Ly6G⁻Ly6C⁺), granulocytic (G-MDSCs; CD11b⁺Ly6G⁺Ly6C^{low/neg}), antigen presenting/dendritic cells (APC/DC; CD11c⁺MHCII⁺) or macrophages (F4/80⁺) (**Extended Data Fig. 8a**). We did observe a slight decrease in DCs in the tumor treated with DT2216 (**Extended Data Fig. 8a**) – though not direct evidence, at least suggesting that the increased antitumor immunity induced by DT2216 is unlikely attributable to the upregulation of DCs to enhance antigen presentation. We could not detect sufficient numbers of Tr1 and Bregs in our tumor models to give a fair comparison because the infiltration of these cells to the tumors studied was too low, suggesting that they are unlikely to play an important role in the enhanced antitumor immunity induced by DT2216.

A setting where Treg cells specifically could not be targeted by the PROTAC would provide the ideal negative control and an excellent test of the author’s hypothesis.

Response: It is a great point. We have given a lot of thought on how to test the hypothesis, but we failed to come up an experimental strategy that is practical at the present. We have DT2216 that is based on VHL E3 ligase to induce BCL-X_L degradation; or PZ15227 that is based on CRBN. Ideally, we can use Treg-specific knockout of VHL and CRBN, which will lead to the unresponsiveness of Tregs to DT2216 and PZ15227 treatment, respectively. Unfortunately, VHL is very important for Treg function, which prohibits the use of those mouse models for our study (2). CRBN germline KO is available but has distinct function in T cell activation and memory (3), which cannot be used to access the role of PZ15227 in Treg effect on T cell activation. Therefore, we are regretful for our inability to address this concern. However, since the tumor cells used in our studies cannot be directly killed by our BCL-X_L degrader and we did not detect any significant changes in other types of immune suppressive cells in the tumor after DT2216 treatment, it is highly likely that the increased antitumor immunity seen after DT2216 treatment is primarily attributable to the depletion of TI-Tregs.

2) *A major strength of the BCL-xL PROTAC is that it avoids the thrombocytopenia induced by pharmacological BCL-xL inhibitors. However, the authors data show that it depletes Treg cells in lymphoid tissues; not just the tumours (Fig 3A). Aside from a perfunctory analysis of three organs, there is no assessment of autoimmunity in the mice treated with the BCL-xL-targeting PROTAC. Is there increased T cell activation? Are there higher circulating autoantibodies in treated mice? What about other tissues? The authors need these data to support their key claim that there are no autoimmune side-effects with this treatment (Abstract, Discussion).*

Response: Thanks for the great suggestions. To address the reviewer's concerns, we did two more experiments: 1) we determined the treatment related CD8 T cell activation in different tissues including lymph nodes, spleens, bloods, lungs, and tumors (**Extended Data Fig. 9a-i**). 2) we used ELISA to analyze serum IgG and IgM, from normal or tumor-bearing mice (**Extended Data Fig. 9j-k**). We used pristine-treated mice as positive control for increased IgG and IgM. Neither experiments show signs of autoimmune onset. This is a supplement to our data in the original manuscript, i.e. no change in tissue histology from most easily attached tissues such as pancreas (pancreatitis, diabetes), lung (asthma, COPD, etc), colon (colitis or other inflammatory bowel diseases) (**Extended Data Fig. 9m**) etc. All these results support DT2216 at the dose treated for cancer therapy does not induce autoimmunity (**Extended Data Fig. 9**). In addition, the safety of DT2216 has been extensively evaluated in three different species (mice, rats and dogs), including GLP toxicity studies, for an IND application, which will be submitted to the FDA at the end of the year to initiate phase I clinical studies in H1 2020. In these studies, no autoimmunity has been observed in the animals. These results are available upon request.

3) *The conclusion that Treg cells infiltrating tumours specifically upregulate BCL-xL, and that other immune cell types do not, is tested broadly, but not very deeply. In most cases, the comparison made is between Treg cells infiltrating the tumour and conventional CD4+ and CD8+ T cells subsets in lymphoid tissues or the blood. CD4+ and CD8+ T cell populations are composed of many functionally distinct subtypes (e.g. naïve, activated, effector, memory) and their representation in lymphoid recirculation and tissues is very different (e.g. many more naïve T cells). Since the expression profiles of BCL-2 family proteins also varies greatly among these subtypes, a more suitable comparison would involve tumour Treg vs. effector/memory CD4+ and CD8+ T cells from lymphoid tissues and non-lymphoid tissues.*

Response: We now include thorough analysis of single RNAseq data from ccRCC (**Extended Data Fig. 3e-g**), as well as protein data for the suggested cell populations (**Extended Data Fig. 3h**). The results from this analysis show that BCL-X_L is exclusively elevated in TI-Tregs.

There is also heterogeneity among Treg cell populations that should also be taken into account. Are all Treg subtypes in tumours sensitive to the PROTAC?

Response: We appreciate the reviewer's insightful comments. In fact, the heterogeneity of human TI-Tregs has been discovered using the same ccRCC and HCC single cell RNAseq datasets in a separate study that is currently under consideration at Nature Communication and that, confidentially, a copy of the related manuscript is made available for their further evaluation of the scRNA seq dataset and analysis. In this additional manuscript, we found that human TI-Tregs have two major branches using pseudotime trajectory analysis (also included as **Extended Data Fig. 1f-g**). We found that BCL-X_L is mainly expressed in Fate-1 Tregs that have the signature of Tregs with superior suppressive functions ($P = 2.48e-08$). We suspect that the Fate-1 TI-Treg cells have elevated BCL-X_L and are likely the primary targets for DT2216, which has been discussed in the revised manuscript (**Page 5, lines 9-20**). We also determined the heterogeneity of TI-Tregs using cell surface markers CD45RA and CCR7, and found that TI-Tregs only consist of CD45RA⁺CCR7⁻ effector memory population (**Extended Data Fig. 2b**). PB-Tregs, on the other hand, exhibit heterogeneous populations (**Extended Data Fig. 2b**). This is in agreement with the observation from Sakaguchi laboratory who initially developed the use of CD45RA for human Treg heterogeneity (4).

Moreover, only 3 patients per tumour type were assayed. Given normal human variation (and among tumours), I am very surprised such tiny standard deviations are reported in Figures 1D, E, G and H. There is also no information on the tumour types or their staging, which makes it harder to discern how broadly this observation applies and how others might reproduce the findings.

Response: We did both technical replicates (3 slices per tumors) and biological replicates (from multiple cancer specimens including 9 PB, 9 breast cancers and summarized (**Fig. 1c-d**, 6 luminal breast cancer and 3 triple negative breast cancers). The data for ccRCC is from one patient with paired PBMC and cancer, showing 3 technical replicates from this patient (**Fig. 1e-f**). We summarized all human cancer and paired PBMC data, including 9 breast cancers, 2 ccRCC, and 2 colon cancers (**Extended Data Fig.3g**). All 13 cancer specimens exhibit BCL-X_L elevation in TI-Tregs relative to Tconv cells or PB-Tregs.

Minor comments:

1) *The data from the FLIVO experiments are not convincing. These data are important because they are the only experiments testing whether apoptosis is the mechanism of Treg cell reduction. The FLIVO shift in Ext. Fig. 4B is modest (at best) and there are no controls guiding the placement of the gate/marker used.*

Response: Thanks for the comment. We included a non-stained control, but FLIVO staining is known to have background staining and we cannot rely on the shift of FLIVO peaks comparing the same non-stained cell populations. We updated new FLIVO shift in **Extended Data Fig.7b**, including unstained control, Tconv cells and Tregs, with or without DT2216 treatment. It is very clear that only the TI-Tregs respond to DT2216 treatment and have a clear shift in the FLIVO signal, reflected by mean fluorescence intensity (MFI in the new **Fig .4b**).

2) *The in vitro cancer slice experiments are innovative, but I am concerned that there would be a lot of basal apoptosis in certain cell types due to hypoxia. This issue might influence the outcomes observed based on the localisation of immune cell types. Further data characterizing this system would be helpful. How long between harvesting the tissue and establishment of the cultures? Also, when samples sizes of 3-4 are quoted, do these refer to the number of slices from an individual donor, or the number of donors assayed.*

Response: We appreciate the note of innovation at our approach. We collected tissues right after surgery and cut them into pieces to establish the cultures without any delay, which was usually done within 3-4 hours after the tumors were removed from the patients. It is possible that these procedures may induce some basal level of cell death due to the hypoxia without blood supply, but we did our best to limit the time of the whole processes and keep every procedures as consistent as possible, such as having the same person to handle all the samples, to reduce variations.

We did both technical replicates (3 slices per tumors) and biological replicates (from multiple cancer specimens including 9 breast cancer, 2 renal and 2 colon cancers, new **Extended Data Fig.3g**). The size and tissue architecture of cancer specimens significantly limited the number of specimens we can get. For other experiments, we were only able to accumulate 3-4 specimens. COVID-19 prohibits our further collection of fresh cancer specimens.

3) *There are no statistical tests applied to the RNAseq data; only expression fold-changes are reported. Some indication of how reproducible or robust these observations are is essential.*

Response: Statistical analysis of these data are now added to **Extended Data Fig. 1 and Extended Data Table 1** in the revised manuscript.

4) *The flow cytometry of BCL-xL expression is important and a strong element in the paper. It would be better if unstained controls were added so readers could assess the extent of the change in MFI and whether the low signals (say, in PB-Treg in Fig 1C) represents low or negative staining. This is an important issue when interpreting potential side-effects of the PROTAC-targeting of BCL-xL.*

Response: Thanks for the suggestion. We included the control staining for the BCL-X_L expression as suggested. The unstained control (NS) is not the best as we showed in **Fig 1c, 1e**. We included a negative control (Neg), using the same tissue pre-treated with DT2216 to deplete BCL-X_L protein (**Fig. 1c, 1e**).

Reviewer #3 (Remarks to the Author): with expertise in Treg and tumor immunology

This manuscript describes the potential application of BCL-XL Proteolysis-Targeting Chimera (PROTAC) as a regulatory T (Treg) cell-targeted reagent. Treg cells play an important role in tumor immunity by suppressing a wide range of antitumor immune responses. Indeed, high frequency of Treg cells and their dominance to effector T cells in the tumor microenvironment (TME) are associated with poor prognosis in various types of cancer. Yet, Treg cell-targeted therapies have not been applied into the clinic, and developing strategies to control Treg cells is an urgent issue in the cancer immunotherapy field. The authors addressed the issue with focusing on high and selective BCL-XL expression by tumor infiltrating Treg cells. BCL-XL was highly expressed by tumor infiltrating Treg cells compared to peripheral Treg cells and other effector T cells such as CD8⁺ T cells and CD4⁺ non-Treg cells in human cancers and murine tumor models. Accordingly, DT2216, the lead BCL-XL PROTAC induced selective reduction of tumor infiltrating Treg cells via apoptosis induction, resulting in the activation of effector T cells. While one major concern with targeting BCL-XL is concomitant reduction of platelets, platelets were not influenced as their DT2216 employed VHL E3 ligase for polyubiquitination and degradation. Furthermore, DT2216 exhibited a strong antitumor efficacy in various tumor models, and the antitumor effects were dependent on CD8⁺ T cells.

General comments;

Given a crucial role of Treg cells in self-tolerance, Treg cell depletion as a whole can trigger autoimmunity in animal models, thereby how Treg cells can be controlled to evoke and augment antitumor immunity

without inducing deleterious autoimmunity, strongly indicating the necessity of developing Treg-cell depletion methods with high selectivity to tumor-infiltrating Treg cells. This paper documents a very interesting approach with their novel Treg cell-targeted reagent. Targeting BCL-X_L for selective Treg cell deletion is a novel concept and their preclinical study data are very promising. Yet, lacking some experiments needs to be revised because of missing important controls (see Specific comments).

Response: We are very grateful to the reviewer's recognition of the significance of the findings reported in our manuscript and his/her insightful recommendations described below.

Specific points;

1. *In Figure 2 and 3, the authors show the selective reduction of Treg cells by apoptosis induction. However, there is a contradictory paper showing that adenosine produced by apoptotic Treg cells induces a strong immune suppression, and macrophage-associated clearance plays an important role for effective immune activation via Treg cell reduction by apoptosis (Maj et al Nat Immunol 18 1332-1341 2017). The authors should address the potential immune suppression by Treg cell apoptosis in their system.*

Response: This is a great point. We have come across this paper from Dr. Zou group while working on the project (5). There is no doubt that TI-Tregs, when undergoing apoptosis, are more immune suppressive due to the release of ATP that can be catabolized into AMP via co-expression of CD39/CD73 on Tregs to conserve the suppressive activity, in addition to the potential release of other immune suppressive molecules such as IL-10, TGF-beta, or others. These molecules, however, work through secretion and mediate Treg function in a contact-independent manner. The majority of published results strongly support a contact-dependent mechanism (6-10). Another issue is the kinetics of apoptotic clearance. Apoptotic clearance by monocyte/macrophages is believed to be an event starting from earlier stages of apoptotic induction (11, 12). Ly6C⁺ monocytes or tumor-associated macrophages (TAM) are much more abundant populations in most cancer models and human cancers than TI-Tregs. The battle between clearance of apoptotic Tregs and maintaining the suppressive functions of Tregs – in our case – favor the clearance. We included a new **Extended Data Fig. 10** to show that caspase activation was detectable at 24 hrs after DT2216 treatment of MC38-tumor bearing mice, but completely disappeared after 48 hrs of treatment, supporting the rapid clearance of these apoptotic TI-Tregs within tumor microenvironment. New discussion was added (page 10, lines 23-23; page 11, lines 1-10).

While we did not see any CD73⁺ TI-Tregs from MC38 tumors, CD39⁺ TI-Tregs exhibit similar percent of cell death comparing vehicle and DT2216 group (**Extended Data Fig. 7g**), suggesting there is no preferential killing of CD39⁺ Tregs by DT2216. Human ccRCC data also showed that there is no CD73 mRNA in TI-Tregs, but with CD39 mRNA expression (*ENTPD1*) (**Extended Data Fig. 1g**). One thing we don't have a good explanation is the replenish of Tregs with as high as 30-50% apoptosis at any given time as shown in Dr. Zou's paper (5), which may be a very specific phenotype in human ovarian cancers.

We have to respectfully point out some experimental differences between our paper and Dr. Zou's paper. We used human kidney, breast, liver, and colon to show the elevated BCL-X_L in TI-Tregs (both mRNA and protein levels), which forms the basis of targeting BCL-X_L to eliminate these cells. This is further confirmed when we used other published datasets to determine the ratio of BCL-X_L expression between Tregs and Tconv cells in breast and several other mouse and human cancers (**Extended Data Fig. 1e**) (13, 14). The phenotype observed by Dr. Zou's group might be more specific to certain groups of human ovarian cancers where BCL-X_L expression is higher in Tconv cells and where tumor microenvironment favors a more apoptotic phenotype of TI-Tregs by an unknown reason.

2. *In Figure 3 and Extended Figure 4, the effects of DT2216 on other immune cells are shown. A previous*

human Treg cell-targeted treatment with anti-CCR4 mAb revealed the complementary expansion of other immune suppressive cells such as MDSCs upon Treg cell depletion (Kurose et al Clin Cancer Res 21 4327-4336 2015). The authors need to show the kinetics of other immune suppressive cells.

Response: This is an outstanding point. We did perform a thorough analysis of all myeloid population and found no significant difference within the proposed immunosuppressive cell types including G-MDSC (CD11b⁺Ly6G⁺Ly6C^{low/-}), M-MDSC (CD11b⁺Ly6G⁺Ly6C^{high}), or tumor-associated macrophages (F4/80⁺). We did not observe any significant difference with DT2216 treatment. The frequency of B220⁺ B cells was very low in our models, which prohibited the analysis of even rarer population of Breg cells in our study. We did see a bit decrease of tumor-infiltrating DC population (CD11c+MHCII+) by DT2216 treatment. All these data are now included in **Extended Data Fig. 8**.

3. In Figure 3 and Extended Figure 7, expansion of entire CD8⁺ T cells upon Treg cell depletion is shown. The authors need to examine tumor antigen-specific CD8⁺ T cell responses.

Response: We greatly appreciate the comments. Based on the data presented in our original **Fig. 3e** (now changed to **Fig. 4e**), we did not see the overall change in CD8 T cell frequency; if any it is slightly decreased in DT2216-treated tumors. We found the significant increase of granzyme B⁺ CD8 T cells in the tumors (**Fig. 4g**), circulation and draining lymph nodes (**Extended Data Fig. 7i**), but not within non-draining lymph nodes or spleen (**Extended Data Fig. 7i**), indicating that the tumor-specific activation of CD8 T cells is likely mediated by DT2216-induced Treg depletion.

In addition, we examined the intrinsic tumor antigen-specific CD8 T cell activation using established neoantigens from MC38 tumor model (15), in response to the reviewer's suggestion. We tried the 3 neoantigens following their previously published protocol but mice receiving these peptides and poly I:C were all dead due to the strong immune reactions, likely due to the different microbiotas altering immune reactions to tumor neoantigens/poly I:C. We tried to reduce the amount of PolyI:C without any luck. The classic OT-1 system/Ova antigen escapes the classic priming stage for CD8 T cells within draining lymph nodes, where Tregs are proposed to play a critical through CTLA-4/CD80 or CD86 binding. Nonetheless, the tumor-specific activation of CD8 T cells, not systematic activation within other tissues, supports the antigen-specific manner and is critical for inhibiting cancer growth mediated by CD8 T cells (**New Fig 3f**). DT-2216 did not induce CD8 T cell activation in non-tumor-bearing mice from various tissues (**Extended Data Fig. 9**), providing a further support for the cancer specific activation of CD8 T cells.

4. DT2216 exhibited a strong antitumor efficacy in various tumor models by depleting Treg cells. Presumably there would be no synergy with other Treg cell depletion methods (anti-CTLA-4 mAb / DTR treatment in FoxP3-DTR mice). By contrast, how about the synergistic effect with PD-1 blockade treatment.

Response: Thanks for the suggestion. We have the same concern regarding the combination with anti-CTLA-4 antibody since both mechanisms are through Treg depletion. We did combine anti-PD-1 antibody (200 ug/mouse every four days using anti-PD-1 is from BioXcell) with DT2216 in Py8119 model. Below

is the tumor growth graph. Anti-PD-1 antibody did not reduce tumor growth in the Py8119 breast cancer model as we knew based on our previous experience, but we were surprised to find that anti-PD-1 antibody cancelled the tumor-suppressing effect of DT2216. We repeated the experiments another time

using 7-8 mice per group and obtained a similar result. We don't know the reason behind the cancellation effect and the underlying mechanism of this finding has yet to be elucidated. Therefore, we did not include the confusing data in the paper since we don't know how to interpret the data.

5. *There are missing control staining in histograms in Figure 1-3 and Extended Figure 2 and 4.*
Response: We have included all negative controls for these histograms.

6. *The references do not appear in order, e.g. 1 followed by 4.*

Response: The MS was transferred from another Nature Research Journal to Nature Communications and the references 2 and 3 were initially cited within the summary section. Now the correct reference order is arranged.

Reviewer #4 (Remarks to the Author): with expertise in scRNA seq analysis

The manuscript "Proteolysis-targeting chimera against BCL-XL destroys tumor-infiltrating regulatory T cells" from Kolb, De and co-authors describe the impact of pharmacological blockage of BCL-XL with a Proteolysis-Targeting Chimera in tumor cell growth. BCL-xl blocking leads to Treg depletion in the tumor and enhanced CD8 T cell response, which leads to reduced tumor growth.

The manuscript makes use of a range of different techniques and models to identify BCL-xl expression in tumor infiltrating lymphocytes and the effects of its pharmacological targeting. The authors analyze a large number of different human and mouse tumor types, which is a very positive aspect of the work, as it offers potential in a broad range of cancer types. This is an impressive and positive aspect of the work, and I have just a few minor comments below.

Response: We greatly appreciate the reviewer's support of our manuscript. The minor comments are addressed below.

1) *To improve readability, due to the large number of different cell lines and cancer models used, I suggest adding in each figure legends the cancer type of the cells at least once. This is present in figures 1 and 2, but not 3 and 4 and many of the supplementary cells.*

Response: Thank you for the suggestion. We have added the information of all cancer types or cell lines in the legends within each figure panel.

2) *The analysis of the single cell RNAseq data generated by the authors is rather superficial, which reduces the re-usability of the dataset. I advise for inclusion of more detailed panels, indicating proper gene markers and better cluster assignments in extended figure 1.*

Response: We appreciate reviewer comments. The tSNE analysis and all detailed information is expected to be published in a separate paper under consideration at Nature Communication and that, confidentially, a copy of the related manuscript is made available for their further evaluation of the scRNA seq dataset and analysis. Therefore, we did not include the information to avoid duplication. We have more sophisticated analysis added to **Extended Data Figs. 1 and 3** as suggested.

3) *Also on supplementary Figure 1, the depiction of % difference versus log-fold change would benefit from a proper statistical measure of difference (p-value).*

Response: We added the statistics for BCL-X_L gene expression data for both ccRCC and HCC as suggested (**Extended Data Fig. 1c-d**).

4) *In the data processing methods parts, the authors write "The data were then log(n+1) transformed" and*

a few sentences later in the same page ", "the average log2 transformed". The correct log conversion should be stated.

Response: Thanks for pointing this out. All bulk RNAseq data were $\log_2(n+1)$ transformed in the manuscript and corrected to reflect it. The Seurat package uses Napierian Log $\ln(n+0.1)$ for all single cell RNAseq analysis. We have made clear in the **Main Text** and the **Methods** sections.

References

1. Khan S, Zhang X, Lv D, Zhang Q, He Y, Zhang P, Liu X, Thummuri D, Yuan Y, Wiegand JS, Pei J, Zhang W, Sharma A, McCurdy CR, Kuruvilla VM, Baran N, Ferrando AA, Kim YM, Rogojina A, Houghton PJ, Huang G, Hromas R, Konopleva M, Zheng G, Zhou D. A selective BCL-XL PROTAC degrader achieves safe and potent antitumor activity. *Nat Med.* 2019;25(12):1938-47. Epub 2019/12/04. doi: 10.1038/s41591-019-0668-z. PubMed PMID: 31792461.
2. Lee JH, Elly C, Park Y, Liu YC. E3 Ubiquitin Ligase VHL Regulates Hypoxia-Inducible Factor-1alpha to Maintain Regulatory T Cell Stability and Suppressive Capacity. *Immunity.* 2015;42(6):1062-74. Epub 2015/06/18. doi: 10.1016/j.immuni.2015.05.016. PubMed PMID: 26084024; PMCID: PMC4498255.
3. Kang JA, Park SH, Jeong SP, Han MH, Lee CR, Lee KM, Kim N, Song MR, Choi M, Ye M, Jung G, Lee WW, Eom SH, Park CS, Park SG. Epigenetic regulation of Kcna3-encoding Kv1.3 potassium channel by cereblon contributes to regulation of CD4+ T-cell activation. *Proc Natl Acad Sci U S A.* 2016;113(31):8771-6. Epub 2016/07/22. doi: 10.1073/pnas.1502166113. PubMed PMID: 27439875; PMCID: PMC4978309.
4. Miyara M, Yoshioka Y, Kitoh A, Shima T, Wing K, Niwa A, Parizot C, Taflin C, Heike T, Valeyre D, Mathian A, Nakahata T, Yamaguchi T, Nomura T, Ono M, Amoura Z, Gorochov G, Sakaguchi S. Functional delineation and differentiation dynamics of human CD4+ T cells expressing the FoxP3 transcription factor. *Immunity.* 2009;30(6):899-911. Epub 2009/05/26. doi: 10.1016/j.immuni.2009.03.019. PubMed PMID: 19464196.
5. Maj T, Wang W, Crespo J, Zhang H, Wang W, Wei S, Zhao L, Vatan L, Shao I, Szeliga W, Lyssiotis C, Liu JR, Kryczek I, Zou W. Oxidative stress controls regulatory T cell apoptosis and suppressor activity and PD-L1-blockade resistance in tumor. *Nat Immunol.* 2017;18(12):1332-41. Epub 2017/10/31. doi: 10.1038/ni.3868. PubMed PMID: 29083399; PMCID: PMC5770150.
6. Takahashi T, Kuniyasu Y, Toda M, Sakaguchi N, Itoh M, Iwata M, Shimizu J, Sakaguchi S. Immunologic self-tolerance maintained by CD25+CD4+ naturally anergic and suppressive T cells: induction of autoimmune disease by breaking their anergic/suppressive state. *Int Immunol.* 1998;10(12):1969-80. Epub 1999/01/14. doi: 10.1093/intimm/10.12.1969. PubMed PMID: 9885918.
7. Thornton AM, Shevach EM. CD4+CD25+ immunoregulatory T cells suppress polyclonal T cell activation in vitro by inhibiting interleukin 2 production. *J Exp Med.* 1998;188(2):287-96. Epub 1998/07/22. doi: 10.1084/jem.188.2.287. PubMed PMID: 9670041; PMCID: PMC2212461.
8. Piccirillo CA, Shevach EM. Cutting edge: control of CD8+ T cell activation by CD4+CD25+ immunoregulatory cells. *J Immunol.* 2001;167(3):1137-40. Epub 2001/07/24. doi: 10.4049/jimmunol.167.3.1137. PubMed PMID: 11466326.
9. Dieckmann D, Plottner H, Berchtold S, Berger T, Schuler G. Ex vivo isolation and characterization of CD4(+)CD25(+) T cells with regulatory properties from human blood. *J Exp*

Med. 2001;193(11):1303-10. Epub 2001/06/08. doi: 10.1084/jem.193.11.1303. PubMed PMID: 11390437; PMCID: PMC2193384.

10. Jonuleit H, Schmitt E, Stassen M, Tuettenberg A, Knop J, Enk AH. Identification and functional characterization of human CD4(+)CD25(+) T cells with regulatory properties isolated from peripheral blood. *J Exp Med*. 2001;193(11):1285-94. Epub 2001/06/08. doi: 10.1084/jem.193.11.1285. PubMed PMID: 11390435; PMCID: PMC2193380.

11. Poon IK, Lucas CD, Rossi AG, Ravichandran KS. Apoptotic cell clearance: basic biology and therapeutic potential. *Nat Rev Immunol*. 2014;14(3):166-80. Epub 2014/02/01. doi: 10.1038/nri3607. PubMed PMID: 24481336; PMCID: PMC4040260.

12. Elliott MR, Chekeni FB, Trampont PC, Lazarowski ER, Kadl A, Walk SF, Park D, Woodson RI, Ostankovich M, Sharma P, Lysiak JJ, Harden TK, Leitinger N, Ravichandran KS. Nucleotides released by apoptotic cells act as a find-me signal to promote phagocytic clearance. *Nature*. 2009;461(7261):282-6. Epub 2009/09/11. doi: 10.1038/nature08296. PubMed PMID: 19741708; PMCID: PMC2851546.

13. Magnuson AM, Kiner E, Ergun A, Park JS, Asinovski N, Ortiz-Lopez A, Kilcoyne A, Paoluzzi-Tomada E, Weissleder R, Mathis D, Benoist C. Identification and validation of a tumor-infiltrating Treg transcriptional signature conserved across species and tumor types. *Proc Natl Acad Sci U S A*. 2018;115(45):E10672-E81. Epub 2018/10/24. doi: 10.1073/pnas.1810580115. PubMed PMID: 30348759; PMCID: PMC6233093.

14. Plitas G, Konopacki C, Wu K, Bos PD, Morrow M, Putintseva EV, Chudakov DM, Rudensky AY. Regulatory T Cells Exhibit Distinct Features in Human Breast Cancer. *Immunity*. 2016;45(5):1122-34. doi: 10.1016/j.immuni.2016.10.032. PubMed PMID: 27851913; PMCID: PMC5134901.

15. Yadav M, Jhunjunwala S, Phung QT, Lupardus P, Tanguay J, Bumbaca S, Franci C, Cheung TK, Fritsche J, Weinschenk T, Modrusan Z, Mellman I, Lill JR, Delamarre L. Predicting immunogenic tumour mutations by combining mass spectrometry and exome sequencing. *Nature*. 2014;515(7528):572-6. Epub 2014/11/28. doi: 10.1038/nature14001. PubMed PMID: 25428506.

REVIEWERS' COMMENTS

Reviewer #1 (not contacted):

Reviewer #2 (Remarks to the Author):

The revised manuscript addressed most of my concerns with new data and clarifications. I commend the authors on this work.

However, there remain two issues related to the central conclusion that the depletion of Treg cells is specific to the tumour, that I suggest bear further consideration.

Specificity of DT2216 for tumour Tregs:

According to the new Figure 4A, Treg cells are at ~0% in 4/5 mice in the spleen and draining lymph node. This represents a dramatic loss of all Tregs in lymphoid tissues. In other models (e.g. Foxp3-DTR) this loss typically leads to auto-inflammatory disease (e.g. PMID: 17136045). Likewise, new Figure 4C shows that 4/5 mice lost all other CD4+ T cells (Tconv) in the draining lymph node. It is surprising that this is not a statistically significant decrease compared to the controls because there is a very similar distribution to the Treg data (which was significantly different). That aside, there are clearly substantial systemic effects on Tregs and conventional CD4+ cells that extend beyond the tumour. These data are not consistent with the overall conclusion, made later, that the depletion of Tregs by DT2216 is specific to the tumour-infiltrating cells. At the very least, this conclusion should be tempered by the observation that the specificity for tumour-infiltrating Treg cells may vary according to individuals or (as in this case) different tumour models.

Measures of autoimmunity:

The authors showed T cell activation data, histology for three organs and total circulating IgM and IgG as evidence that there is no autoimmunity. (Note that I believe there is a typo; "pristine" should probably be "pristine"). These are relatively gross measures, mainly focused on the possibility of SLE-like disease. Treg deficiency tends to cause different manifestations that are not examined (e.g. gut, liver, skin pathology, lymphadenopathy, e.g. PMID: 17136045). In their response, the authors refer to a more extensive analysis of autoimmunity in mice as pre-clinical data for their drugs. I'd suggest these data should be added to this manuscript to support the author's conclusion that there is no autoimmunity when Treg are targeted with DT2216 (esp. in light of the observations regarding the severe Treg cell depletion observed in the MC38 model).

Reviewer #3 (Remarks to the Author):

The manuscript by Kolb. R. et al. has been revised extensively and addresses most of the reviewer's concerns. I congratulate the authors for the excellent job in revising the manuscript.

Reviewer #4 (Remarks to the Author):

The authors addressed most of my comments. The data analysis is better described and more reproducible at this stage.

I only have one minor comment related to the point below

"3) Also on supplementary Figure 1, the depiction of % difference versus log-fold change would benefit from

a proper statistical measure of difference (p-value).

Response: We added the statistics for BCL-XL gene expression data for both ccRCC and HCC as

suggested
(Extended Data Fig. 1c-d)."

I do see the the authors included an adjusted pvalue for the genes used later in the paper, but what the colors (blue and orange) in these figures cannot be found in the legend.

Felipe A Vieira Braga

Point-by-point responses to the reviewers' comments

We appreciate all the reviewers' comments and positive feedback.

Reviewer #2 (Remarks to the Author):

The revised manuscript addressed most of my concerns with new data and clarifications. I commend the authors on this work.

However, there remain two issues related to the central conclusion that the depletion of Treg cells is specific to the tumour, that I suggest bear further consideration.

Specificity of DT2216 for tumour Tregs:

According to the new Figure 4A, Treg cells are at ~0% in 4/5 mice in the spleen and draining lymph node. This represents a dramatic loss of all Tregs in lymphoid tissues. In other models (e.g. Foxp3-DTR) this loss typically leads to auto-inflammatory disease (e.g. PMID: 17136045). Likewise, new Figure 4C shows that 4/5 mice lost all other CD4+ T cells (Tconv) in the draining lymph node. It is surprising that this is not a statistically significant decrease compared to the controls because there is a very similar distribution to the Treg data (which was significantly different). That aside, there are clearly substantial systemic effects on Tregs and conventional CD4+ cells that extend beyond the tumour. These data are not consistent with the overall conclusion, made later, that the depletion of Tregs by DT2216 is specific to the tumour-infiltrating cells. At the very least, this conclusion should be tempered by the observation that the specificity for tumour-infiltrating Treg cells may vary according to individuals or (as in this case) different tumour models.

Response: We agree to the reviewer's suggestion. We tuned down the conclusion that DT2216 can specifically deplete TI-Treg and described data to indicate that DT2216 may selectively deplete TI-Treg in a tumor/individual mouse-dependent manner in the revised manuscript (page 10, lines 212-213) as suggested by the reviewer because in normal mice (non-tumor bearing mice) DT2216 treatment had not significant effect on the percentage of live Treg in the blood, spleen, lymph nodes and lung but slightly increased that in the thymus, suggesting that thymic Treg development doesn't require BCL-XL for survival (Supplementary Figure 9).

Measures of autoimmunity:

The authors showed T cell activation data, histology for three organs and total circulating IgM and IgG as evidence that there is no autoimmunity. (Note that I believe there is a typo; "pristine" should probably be "pristane"). These are relatively gross measures, mainly focused on the possibility of SLE-like disease. Treg deficiency tends to cause different manifestations that are not examined (e.g. gut, liver, skin pathology, lymphadenopathy, e.g. PMID: 17136045). In their response, the authors refer to a more extensive analysis of autoimmunity in mice as pre-clinical data for their drugs. I'd suggest these data should be added to this manuscript to support the author's conclusion that there is no autoimmunity when Treg are targeted with DT2216 (esp. in light of the observations regarding the severe Treg cell depletion observed in the MC38 model).

Response: We greatly appreciate the reviewer's insightful comments. We cited the seminar work from Kim JM et al. (Kim JM et al. Nat Immunol. 8:191, 2007) in our revised manuscript. However, unlike *Foxp3*⁻ mice, we did not observe any significant changes in the size of lymph nodes and the spleens in DT2216-treated mice nor significant expansion of lymphocytes in these tissues (Supplementary Figure 9). In addition, no significant lymphocyte infiltration and tissue abnormality were observed in the lungs, pancreas and intestine from the mice treated with D2216. No induction of autoimmunity was reported in mice that were treated with A-1331852, a selective Bcl-xl inhibitor (Leverson JD et al. Sci Transl Med. 7: 279ra40, 2015), suggesting that Bcl-xl inhibition might be different from inhibition of CTLA-4 and knockout of *Foxp3* in terms of induction of autoimmunity. However, we agree with the reviewer that further studies are needed to examine whether DT2216 treatment may cause any autoimmune pathological changes in the liver, skin and other organs, which has been added to the revised manuscript (page 10, lines 223-225). These studies are ongoing efforts by the Dialectic Therapeutics (<https://www.dtsciences.com/>), which develops DT2216 for the IND application and clinical studies in early 2021.

Reviewer #4 (Remarks to the Author):

"3) Also on supplementary Figure 1, the depiction of % difference versus log-fold change would benefit from a proper statistical measure of difference (p-value). I do see the authors included an adjusted pvalue for the genes used later in the paper, but what the colors (blue and orange) in these figures cannot be found in the legend.

Response: We thank the reviewer for his/her suggestions. We added the statistics for BCL-XL gene expression data for both ccRCC and HCC as suggested (Extended Data Fig. 1c-d); we also updated the legend to define the blue and red.